# RIF1 integrates DNA repair and transcriptional requirements during the establishment of humoral immune responses

Eleni Kabrani [1,8], Ali Rahjouei[1,6,8], Maria Berruezo-Llacuna[1,2], Svenja Ebeling[1,3], Tannishtha Saha[1,3], Robert Altwasser[1,7], Veronica Delgado-Benito [1], Rushad Pavri [4] & Michela Di Virgilio [1,5] ✉

The establishment of protective immune responses relies on the ability of terminally differentiated B cells to secrete a broad variety of antigen-specific antibodies with different effector functions. RIF1 is a multifunctional protein that promotes antibody isotype diversification via its DNA end protection activity during class switch recombination. In this study, we showed that RIF1 ablation resulted in increased plasmablast formation ex vivo and enhanced terminal differentiation into plasma cells upon immunization. Mechanistically, this phenotype is independent from RIF1's role in DNA repair and class switch recombination, and reflects its ability to modulate the transcriptional status of a subset of BLIMP1 target genes. Therefore, here we show that, in addition to promoting antibody diversification, RIF1 fine-tunes the kinetics of late B cell differentiation, thus providing an additional layer of control in the establishment of humoral immunity.

During early B cell development in the bone marrow, common lymphoid progenitors develop into immature B cells via a step-wise process linked to the rearrangement of their antibody receptor/immunoglobulin (Ig) genes by V(D)J recombination[1,2]. Immature B cells expressing a functional and non-autoreactive B cell receptor (BCR) migrate to the periphery, where they complete their maturation into quiescent resting cells[1,2]. The encounter of these naïve mature B cells with their cognate antigen in secondary lymphoid organs triggers late B cell differentiation into effector cells, often accompanied by further BCR/Ig diversification via somatic hypermutation (SHM) and class switch recombination (CSR)[3]. SHM and CSR provide the molecular bases for generating high-affinity and isotype-switched BCRs/Igs,

respectively[4]. At the cellular level, the combined effect of BCR engagement, helper T and follicular dendritic cell-derived signals induces the formation of specialized microstructures known as germinal centers (GCs), where activated B cells undergo a major proliferative burst, selection of high-affinity BCR variants, and differentiation into either antibody-secreting cells (ASCs, plasmablasts (PBs) and plasma cells (PCs)) or memory B cells[5]. B cell activation can also lead to PB generation at extrafollicular sites, which are also characterized by B cell proliferation and represent the main source of ASCs in T-cell independent responses[6]. BLIMP1 (encoded by the *Prdm1* gene) is the transcriptional master regulator of late B cell differentiation[7]. BLIMP1 is dispensable for the initiation of the ASC program but is

[1]Laboratory of Genome Diversification & Integrity, Max Delbrück Center for Molecular Medicine in the Helmholtz Association, 13125 Berlin, Germany. [2]Humboldt-Universität zu Berlin, 10117 Berlin, Germany. [3]Freie Universität Berlin, 14195 Berlin, Germany. [4]Peter Gorer Department of Immunobiology, School of Immunology & Microbial Sciences, King's College London, London SE1 9RT, UK. [5]Charité-Universitätsmedizin Berlin, 10117 Berlin, Germany. [6]Present address: Department of Anesthesiology and Intensive Care Medicine, and Experimental and Clinical Research Center, Charité-Universitätsmedizin Berlin, 10117 Berlin, Germany. [7]Present address: Department of Hematology, Oncology, and Cancer Immunology, Charité-Universitätsmedizin Berlin, 10117 Berlin, Germany. [8]These authors contributed equally: Eleni Kabrani, Ali Rahjouei. ✉e-mail: michela.divirgilio@mdc-berlin.de

necessary for the establishment of the terminally differentiated PC phenotype, mainly through suppression of its target genes[8–11]. The generation of PBs and long-lived PCs capable of secreting high-affinity Igs of different classes provides the foundation for the establishment of protective humoral responses.

RIF1 (Rap1-Interacting Factor 1 Homolog / Replication Timing Regulatory Factor 1) is a multifunctional protein initially identified in budding yeast as a regulator of telomere length homeostasis[12]. In mammalian cells, RIF1 contributes to preserving genome stability during both DNA replication and repair. Under conditions of DNA replication stress, RIF1 protects nascent DNA at stalled forks from degradation, facilitating their timely restart[13–16]. During the repair of DNA double-strand breaks (DSBs), RIF1 participates in the 53BP1-dependent cascade that protects the broken DNA ends against nucleolytic resection, thus influencing the choice of which DSB repair pathway to engage[17–21]. Specifically, RIF1 localizes to DSB-containing chromatin via its interaction with phosphorylated 53BP1, and it counteracts DSB end resection by recruiting the downstream protein complexes MAD2L2-SHLD1-SHLD2-SHLD3 (Shieldin) and CTC1-STN1-TEN1 (CST)[17–30]. In addition to these genome-protective functions, RIF1 plays a central role in the control of DNA replication timing programs in both yeast and higher eukaryotes[31–34]. Several studies have also implicated RIF1 in early mouse development[35–39]. This role appears independent from RIF1's various activities in DNA metabolism, and is mediated by its ability to modulate the transcriptional networks responsible for embryonic stem cell state stability and differentiation[35–39].

Isotype diversification by CSR occurs via a deletional recombination reaction at the Ig heavy chain (*Igh*) locus, which replaces the constant (C) region of the IgM/IgD basal isotype with one of the downstream C regions encoding for the different classes (IgG, IgE or IgA)[4]. The process is initiated by the B cell-specific enzyme Activation-Induced Deaminase (AID), which targets highly repetitive regions preceding the C genes, known as Switch (S) regions[40]. AID deaminates cytosine residues within single-stranded DNA stretches that are uncovered by non-coding transcription across the recombining S-C units[40]. The resulting U:G mismatches are converted into multiple DSBs via the intervention of the base excision and mismatch repair pathways[4]. In the resolution phase, paired DSBs at the donor Sμ and acceptor Sx regions are ligated primarily via the nonhomologous end-joining (NHEJ) DSB repair pathway[41]. This ligation step not only re-establishes the locus integrity but also places the new C region close to the VDJ exon, thus completing CSR at the DNA level[4].

Repair by end-joining is inhibited by extensive resection of S region breaks[4,41]. As a consequence, CSR is heavily dependent on the 53BP1-RIF1-Shieldin-CST machinery[17–19,22–26,30,33,42–47]. In this study, we report an additional role for RIF1 in the regulation of humoral immunity that is independent from its DSB end protection function. We discovered that RIF1 expression is upregulated in mature B cells following activation, where it binds cis-regulatory elements of genes involved in B cell function and differentiation. RIF1 deficiency skews the transcriptional profile of activated B cells towards PBs and PCs, and is associated with an accelerated differentiation to ASCs both *ex* and in vivo. RIF1 directly binds several BLIMP1 target genes and counteracts their premature repression. Thus, RIF1 contributes an additional regulatory layer to the B cell differentiation program that is essential to establish secreted antibody diversity.

## Results

### *Rif1* expression is regulated during B cell differentiation
Given RIF1's multiple roles during early development and in differentiated somatic cells[35,36,38,39,48,49], we asked whether RIF1 contributes functions beyond the regulation of DNA end processing and repair in B cells. To do so, we first monitored *Rif1* expression levels across B cell lineage developmental stages using the Immunological Genome

Project (ImmGen) transcriptomics data[50]. *Rif1* expression varied considerably in the different B cell subtypes, with GC cells exhibiting the highest levels (Fig. 1a). In contrast, the expression of RIF1's up- and downstream partners in DNA end protection, 53BP1 and the Shieldin (MAD2L2-SHLD1-SHLD2-SHLD3) complex, respectively, did not show any major changes during B cell development and differentiation (Fig. 1a and Supplementary Fig. 1a). In addition, ex vivo activation of isolated naïve B cells (Fig. 1b and Supplementary Fig. 1b) resulted in upregulation of *Rif1* transcript levels, regardless of the stimulation condition (LPS and IL-4 (LI); LPS, BAFF and TGFβ (LBT); or LPS (L)) and resulting isotype switching (IgG1, IgG2b, or IgG3, respectively) (Fig. 1c). MAD2L2 was the only additional component of the 53BP1-RIF1-Shieldin axis to exhibit transcriptional induction after ex vivo stimulation (Fig. 1c and Supplementary Fig. 1c). Finally, *Rif1* transcripts' upregulation was accompanied by an increase in the expression also at the protein level, which was readily detectable at 48 h after activation, sustained at 72 h and declining thereafter (Fig. 1d). We concluded that *Rif1* expression is induced following activation of mature B cells both *in* and ex vivo, and that this expression dynamics does not reflect a general behavior of DSB end protection factors.

### RIF1 limits the ex vivo differentiation of activated B cells into PBs and PC-like cells
The specific upregulation of RIF1 expression in GC and ex vivo stimulated splenocytes (Fig. 1) suggests that RIF1's functional repertoire in activated B cells might extend beyond its established role in the regulation of DNA end processing. In addition to SHM and CSR, activation of mature B cells triggers the late stages of B cell development[5,6,51]. Therefore, we next investigated the potential involvement of RIF1 during ASC generation.

Stimulation of naïve B cells with LPS and IL-4 induces their ex vivo differentiation into PBs (evidenced by the surface expression of CD138). To assess the contribution of RIF1 to this process, we employed B lymphocyte cultures from *Rif1^{F/F}Cd19^{Cre/+}* mice (Fig. 1d), which conditionally ablate *Rif1* expression during early B cell development (Supplementary Fig. 2 and[18,52]). We found an over twofold increase in the percentage of CD138+ cells in *Rif1^{F/F}Cd19^{Cre/+}* samples compared to *Cd19^{Cre/+}* (Fig. 2a), despite similar proliferation rates between the two genotypes (Fig. 2b and[18]). Furthermore, whereas controls exhibited a steep rise in PB formation mainly at the last cell division, RIF1-deficient cultures showed higher and increasing levels of PBs from earlier division rounds (Fig. 2b).

To assess whether the PB phenotype that we observed in *Rif1^{F/F}Cd19^{Cre/+}* cultures is a consequence of their deregulated DSB end processing, we analysed B cells from *Shld1^{-/-}* mice[45]. Deficiency in any of the Shieldin subunits abrogates RIF1-dependent DSB end protection, thus severely impairing CSR ([22–26,42–45] and Fig. 2c). We found that *Shld1^{-/-}* cultures displayed levels of PBs comparable to control samples, indicating that the increased PB formation in *Rif1^{F/F}Cd19^{Cre/+}* B cells is independent from RIF1 DSB end protection activity (Fig. 2c). Furthermore, the observed phenotype is not linked to the CSR defect per se since B cell cultures from mice lacking AID (*Aicda^{-/-}*) differentiated into PBs to the same extent as controls (Fig. 2c).

Currently, no in vitro setting can fully recapitulate the complexity of PC differentiation and function. However, the induced GC B (iGB) culture system ideated by Nojima *et al.*[53] mimics the T cell-dependent generation of GC B cells and enables the in vitro manipulation of their fates into either memory- or long-lived PC-like cells[53] (Fig. 2d). We took advantage of this system to assess the contribution of RIF1 to the terminal differentiation of activated B cells ex vivo. At the GC-like phenotype stage (four days stimulation with IL-4 on CD40L- and BAFF-expressing feeder cells), RIF1 deficiency resulted in the expected severe defect in CSR (Fig. 2d, e, and[17–19]). However, and in agreement with their increased potential to differentiate into PBs (Fig. 2a–c),

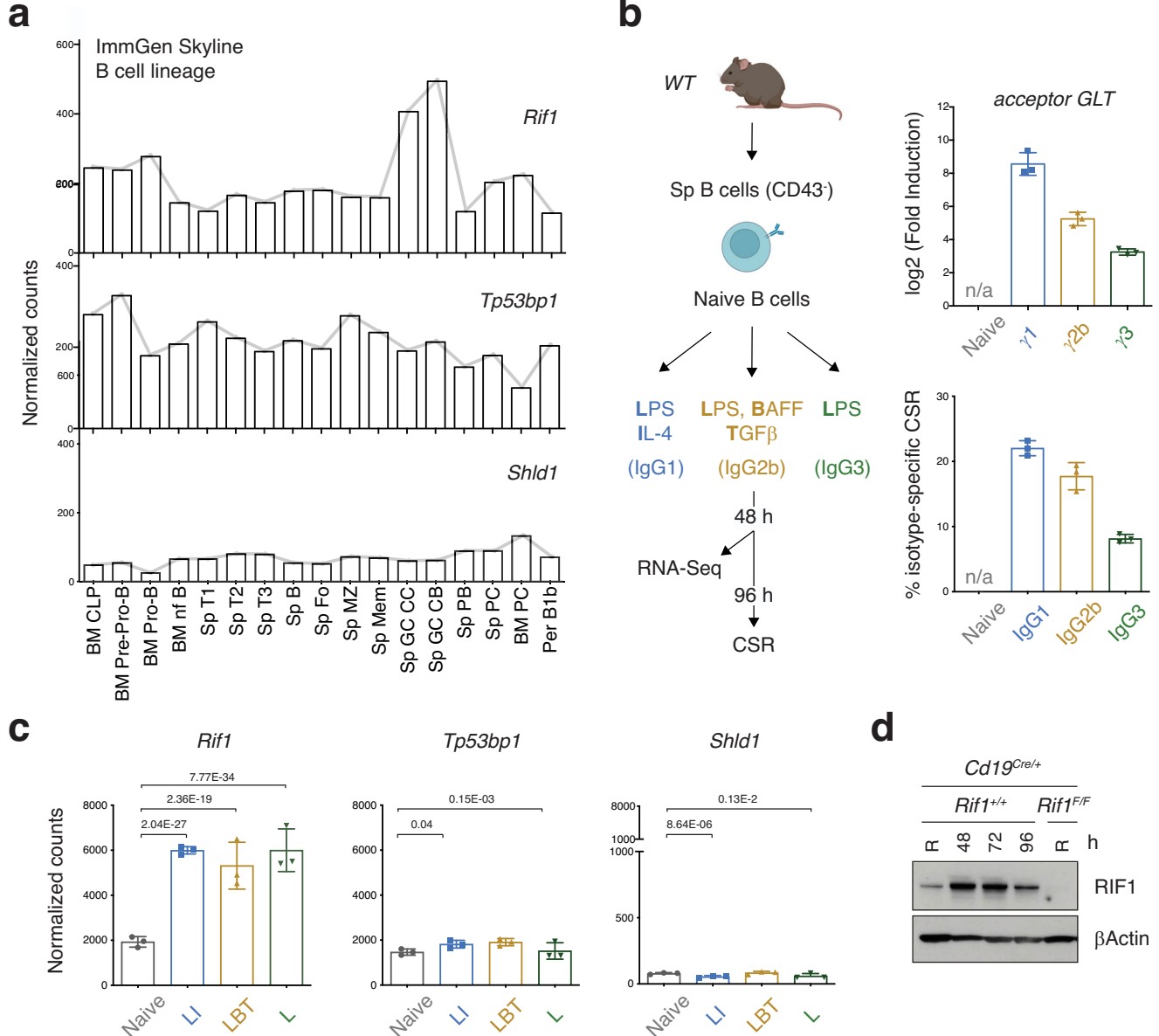

**Fig. 1 | *Rif1* expression is dynamically regulated during late B cell differentiation. a** Expression of *Rif1*, *TpS3bp1*, and *Shld1* genes across B cell lineage developmental stages as determined by the Immunological Genome Project (ImmGen) Skyline RNA-Seq analysis. BM: bone marrow; CLP: common lymphoid progenitor; nf: newly formed; Sp: splenic; Fo: follicular; MZ: marginal zone; Mem: memory; GC: germinal center; CC: centrocytes; CB: centroblasts; PB: plasmablasts; PC: plasma cells; Per B1b: peritoneal B1b. **b** Left: Schematic representation of gene expression analysis in naïve B cells isolated from mouse spleens and stimulated ex vivo with LPS and IL-4 (LI cocktail), LPS, BAFF and TGFβ (LBT), or LPS only (L). Right: Levels of acceptor germline transcript (GLT) induced by LI-, LBT- or L-stimulation (expressed as fold increase over the naïve sample) and CSR efficiency to the corresponding isotype for each of the primary B cell cultures employed in the RNA-Seq analysis (n

= three mice per stimulation condition). n/a: not applicable. **c** Expression of *Rif1*, *TpS3bp1*, and *Shld1* in naïve and LI/LBT/L-stimulated B cells. **d** Western blot analysis of *Cd19^{Cre/+}* and *Rif1^{F/F} Cd19^{Cre/+}* purified B lymphocytes at the indicated times after activation with LI. Data is representative of two independent genotype pairs. R: Resting. Expression values in panels (**a**, **b** and **c**) were normalized by DESeq2, and are presented as mean values ± SD in panels b and c. Significance in panel (**c**) was calculated using a two-sided Wald test, with adjustments for multiple comparisons made through the Benjamini-Hochberg procedure. The adjusted *p* value of significant differences between samples is indicated. Source data are provided as a Source Data file. Created in BioRender. Di virgilio, M. (2025) BioRender.com/o01z838.

---

*Rif1^{F/F}Cd19^{Cre/+}* B cells showed a near twofold increase in the percentage of PC-like cells after prolonged culturing in the presence of IL-21 (Fig. 2,d, f). This finding is particularly striking if we take into consideration that IgM⁺CD138⁺ cells were eventually outcompeted by the switched counterparts in both genotypes (Fig. 2g).

Altogether, these data indicate that RIF1 deficiency accelerates the ex vivo differentiation of activated B cells to PBs and PC-like cells, and that this function is independent from RIF1's role in DNA end protection and CSR.

## RIF1 deficiency does not affect GC B cell dynamics
We next asked whether RIF1 is able to modulate late B cell differentiation also in vivo. To this end, we first assessed the consequences of RIF1 ablation on GC physiology by comparing GC B cell dynamics in Peyer's patches (PPs) of *Cd19^{Cre/+}* and *Rif1^{F/F}Cd19^{Cre/+}* mice. GCs are anatomically separated into the dark zone (DZ) and light zone (LZ)[5,6], and cycling of positively selected GC B cells between LZ and DZ ensures further affinity maturation and clonal expansion[5,6]. PPs are specialized secondary lymphoid structures that line the wall of the

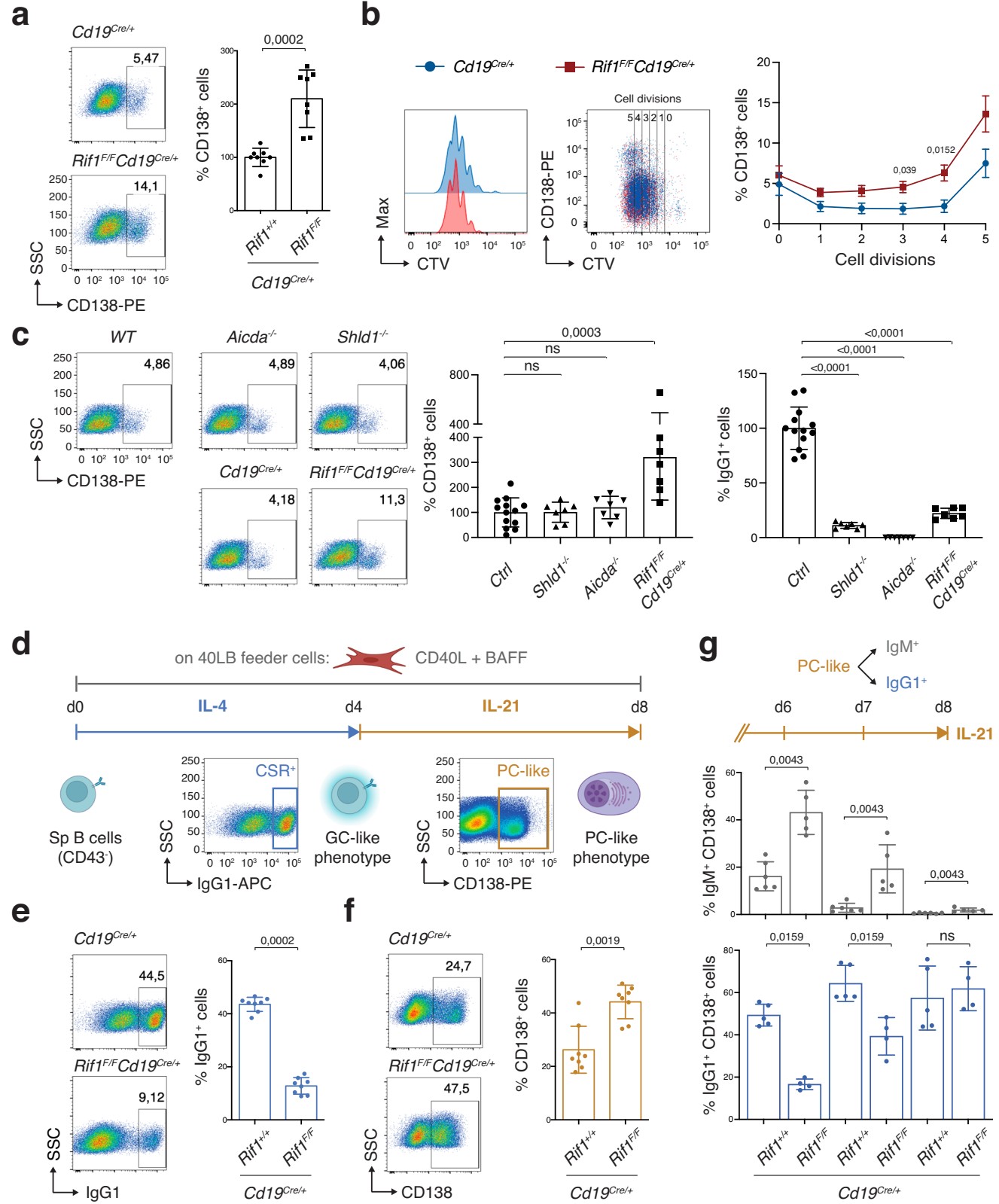

small intestine and are crucial for mucosal IgA antibody responses[54]. PPs allow for the assessment of chronic GC activity in response to a variety of food- and microbiome-derived antigens.

Control and *Rif1^{F/F}Cd19^{Cre/+}* mice showed no significant difference in the percentage of PP GC B cells nor in their distribution in the LZ versus DZ (Fig. 3a, b). In contrast, mice knock-out for AID exhibited the expected massive increase in GC B cells and a cellular distribution

heavily skewed towards the LZ (Fig. 3b and[55–57]). Interestingly, despite the severe CSR defect caused by RIF1 deficiency[17–19,41], PPs of *Rif1^{F/F}Cd19^{Cre/+}* mice displayed near-physiological levels of IgA^+ GC B cells (Fig. 3a, b), thus indicating that the few cells that successfully class-switch are actively selected forward in this context. We concluded that under conditions of chronic stimulation, RIF1 is dispensable for GC B cell formation, selection and proper DZ/LZ compartmentalization.

**Fig. 2 | RIF1 limits the ex vivo differentiation of activated B cells to PBs and PC-like cells independently from its role in DNA end protection and CSR. a** Left: Representative flow cytometry plots measuring percentage of plasmablasts (CD138 + ) in purified splenic B cell cultures 96 h after activation with LPS and IL-4. Right: Summary graph for five independent experiments (n = eight mice per genotype), with values within each experiment normalized to the average of controls, which was set to 100%. **b** Left: Representative plots measuring proliferation by CellTrace Violet (CTV) dilution and percentage of CD138+ cells per cell division. Right: Summary graph for three independent experiments (*n* = six mice per genotype). **c** Left: Representative plots measuring percentage of CD138+ cells and summary graph for four independent experiments (*n* = thirteen (control) and seven mice (for the respective genotypes)). Right: Graph summarizing percentage of IgG1+ cells in the same cultures. CD138+ and CSR values within each experiment were normalized to the average of control mice (*WT* and *Cd19^{Cre/+}: Ctrl*), which was set to 100% (n = thirteen (control) and seven mice (for the respective genotypes)).

**d** Schematic representation of the iGB cell culture system for the ex vivo differentiation of splenic naïve B cells to GC- and plasma cell (PC) -like cells. CD40L: CD40 ligand; d: day; Sp: splenic. Left: Representative plots measuring CSR to IgG1 (**e**) and percentage of plasmablasts (**f**) at day 4 (**e**) and 8 (**f**), respectively, of the iGB cell culture. Right: Summary graphs for five independent experiments (n = eight mice per genotype). **g** Graphs depicting the percentage of IgM+ and IgG1+ cells in the CD138+ populations at the indicated timepoints of the iGB cell culture. n = six *Cd19^{Cre/+}* mice for IgM, five *Rif1^{F/F}Cd19^{Cre/+}* for IgM, five *Cd19^{Cre/+}* for IgG1, and four *Rif1^{F/F}Cd19^{Cre/+}* for IgG1 in three independent experiments. Significance in panels (**a**, **b**, **c**, **e**, **f**, **g**) was calculated with a two-sided Mann–Whitney U test, with data presented as mean values ± SD in panels (**a**, **c**, **e**, **f**, and **g**), and SEM in panel (**b**). Only significant differences were indicated in panel (**b**). ns: not significant. Source data are provided as a Source Data file. Created in BioRender. Di virgilio, M. (2025) BioRender.com/f53q006.

## RIF1 curtails plasma cell formation following immunization

Next, we analyzed the GC and PC compartment of control and *Rif1^{F/F}Cd19^{Cre/+}* mice immunized with the T-cell-dependent antigen 4-Hydroxy-3-nitrophenylacetyl hapten conjugated to Chicken Gamma Globulin (NP-CGG) (Fig. 4a). The total spleen and bone marrow cellularity was not affected by RIF1 deficiency (Supplementary Fig. 3a). However, and in contrast to chronic stimulation, NP-CGG immunization elicited a reduced number of GC B cells in the absence of RIF1 (Supplementary Fig. 3b, c). This phenotype is likely due to higher levels of apoptosis induced by unrepaired DSBs, as we have previously shown[18], or DNA end resection-dependent BCR loss[58]. As expected for an acute immune response context, the levels of class-switched IgG1+ GC B cells (CD19+B220+FAS^{high}CD38^{low}IgG1+) were decreased in *Rif1^{F/F}Cd19^{Cre/+}* mice (Supplementary Fig. 3b, c).

Despite the reduced number of GC B cells, *Rif1^{F/F}Cd19^{Cre/+}* mice displayed a consistent increase in the total pool of terminally differentiated PCs (TACI+CD138+AA4.1+MHCII-) compared to controls in both spleen and bone marrow at earlier time points post-immunization (median PC increase over control levels of 50% and 39% at day 7 and 14, respectively, in the spleen, and of 17% at day 14 in the bone marrow) (Fig. 4b). The higher proportion of PCs observed at day 7 in the spleen occurs on the background of a reduced ASC pool (Fig. 4b), which suggests that terminal B cell differentiation is accelerated in the absence of RIF1 also in vivo. The phenotype was no longer observed in either compartment at later time points (day 28, Fig. 4b) nor in unimmunized mice (Supplementary Fig. 4 and[17]). However, functional assessment of NP-specific PCs by ELISpot analysis showed a sustained trend towards higher levels of ASCs in the bone marrow of *Rif1^{F/F}Cd19^{Cre/+}* mice compared to controls, which was particularly pronounced at later time points after immunization (Fig. 4c). This latter observation is in agreement with the ability of terminally differentiated PCs to home to the bone marrow.

These findings indicate that RIF1 deletion removes a physiological restraint imposed over the terminal differentiation of mature B cells in vivo, which, though concealed under steady state conditions, is readily detectable at the systemic level upon immunization. Furthermore, taking into consideration also the ex vivo recapitulation of the increased ASC phenotype (Fig. 2) and unaffected GC B cell dynamics (Fig. 3), we concluded that RIF1's ability to delay terminal B cell differentiation in vivo reflects a cell-intrinsic property independent from the GC reaction.

## RIF1 deficiency skews the transcriptional profile of activated B cells towards ASCs

RIF1 has been implicated in the modulation of transcriptional programs responsible for embryonic stem cell differentiation[36,39,48,49]. Therefore, we asked whether the increased potential for terminal B cell differentiation in the absence of RIF1 reflects a transcriptional role in the modulation of B cell identity.

Activation of isolated splenic B cells with specific stimuli recapitulates several features of terminal B cell differentiation (Fig. 2 and[10,59]). Hence, we monitored the consequences of RIF1 deficiency on the mature B cell transcriptome after ex vivo activation with LPS and IL-4. Comparative assessment of the transcriptional profiles of *Cd19^{Cre/+}* and *Rif1^{F/F}Cd19^{Cre/+}* B cells (Fig. 5a) identified a large number of genes that were significantly (adjusted *p* value <= 0.05) up- or down- regulated in the absence of RIF1 (149, 732, 672 up- and 230, 662, 981 down-regulated at 48, 72, and 96 h post-activation, respectively) (Fig. 5b, Supplementary Fig. 5a and b, and Supplementary Data 1). However, only a limited number of them exhibited considerably deregulated expression levels in *Rif1^{F/F}Cd19^{Cre/+} versus Cd19^{Cre/+}* cultures (N° genes with log2 FC < -1 and > 1 = 0, 105, and 47 at 48, 72, and 96 h, respectively) (Fig. 5b and Supplementary Data 1). Furthermore, the expression of the mature B cell identity transcriptional regulators (*Pax5, Ebf1, Foxo1*, and *Bach2*) was not affected (Fig. 5c and Supplementary Data 1). However, when we assessed the status of the key factors driving the ASC program (*Prdm1, Irf4*, and *Xbp1*), we found a near twofold increase in *Prdm1* transcript levels at 96 h after activation (Fig. 5b, c). Since the *Prdm1*-encoded transcription factor BLIMP1 is required for the differentiation of pre-PBs into PBs and PCs[8,10], we cross-referenced the list of differentially expressed genes (DEG, adjusted *p* value < 0.05) from *Rif1^{F/F}Cd19^{Cre/+}* activated B cells (Supplementary Data 1) with the PB and PC signatures[10] (see Data analysis section of Methods). We observed a significant overlap between DEGs and the corresponding up-/down-regulated gene set in the ASC signatures, with the tendency being more pronounced for the down-regulated datasets (Fig. 5d, Supplementary Fig. 5c and Supplementary Data 2). We concluded that RIF1 deficiency in activated B cells results in a deregulated expression profile enriched in genes normally expressed in terminally differentiated B cells.

## RIF1 binds genes involved in the modulation of adaptive immune responses

To uncover the molecular mechanism underlying RIF1's ability to maintain the transcriptional identity of mature B cells after activation, we monitored the genome-wide occupancy of RIF1 in activated B cells from *Rif1^{FH/FH}* mice under the same stimulation conditions employed for the ex vivo transcriptional analyses (Fig. 6a,[34]). *Rif1^{FH/FH}* splenocytes express physiological levels of a knock-in 1×Flag-2×Hemagglutinin-tagged version of RIF1 (RIF1^{FH},[60]) that supports its roles in mouse embryonic fibroblasts, embryonic stem cells, and B cells[18,32,60]. We found that the vast majority of RIF1 peaks colocalized with promoters (56,4 %) and distal intergenic regions (25,4 %) (Fig. 6b and Supplementary Data 3). We next performed a functional enrichment analysis via Genomic Regions Enrichment of Annotations Tool (GREAT)[61] to assess the functional significance of both proximal- and distal-to-gene RIF1-binding events. We identified several categories of genes associated with the regulation of

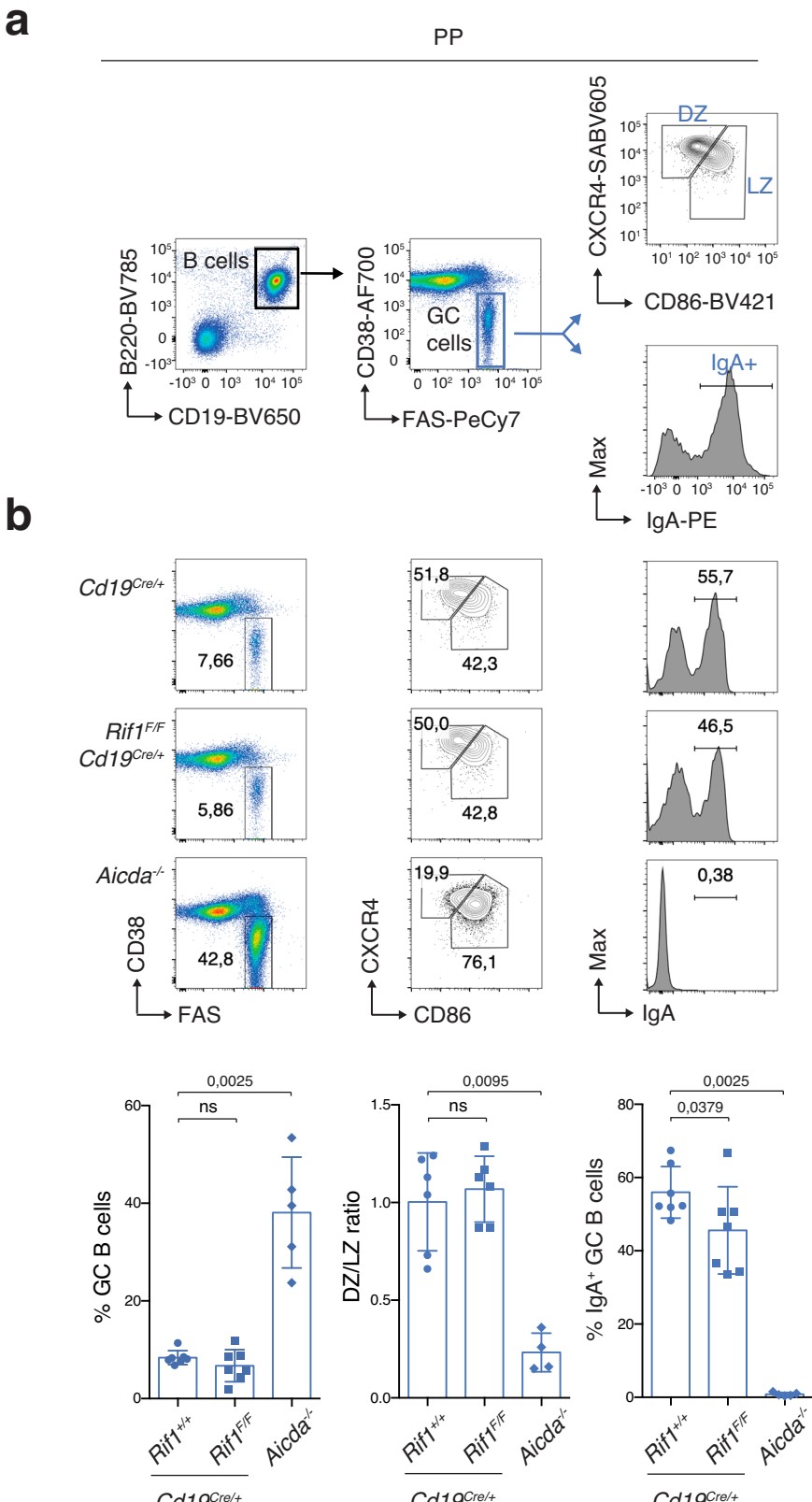

**Fig. 3 | RIF1 deficiency does not affect GC B cell dynamics. a** Gating strategy for Germinal Center (GC) analysis in Payer's Patches (PPs). DZ: dark zone; LZ: light zone. **b** Top: Representative flow cytometry plots measuring percentage of GC B cells, DZ/LZ and IgA+ fraction of GC B cells in PPs of *Cd19*^Cre/+, *Rif1*^F/F*Cd19*^Cre/+ and *Aicda*^-/- mice. Bottom: Summary graphs for n = seven *Cd19*^Cre/+, seven *Rif1*^F/F*Cd19*^Cre/+ and five *Aicda*^-/- mice for GC and IgA+ fractions, and six *Cd19*^Cre/+, six *Rif1*^F/F*Cd19*^Cre/+ and four *Aicda*^-/- mice for DZ/LZ in at least three independent experiments. Significance in panel (**b**) was calculated with a two-sided Mann–Whitney U test, with data presented as mean values ± SD. ns: not significant. Source data are provided as a Source Data file.

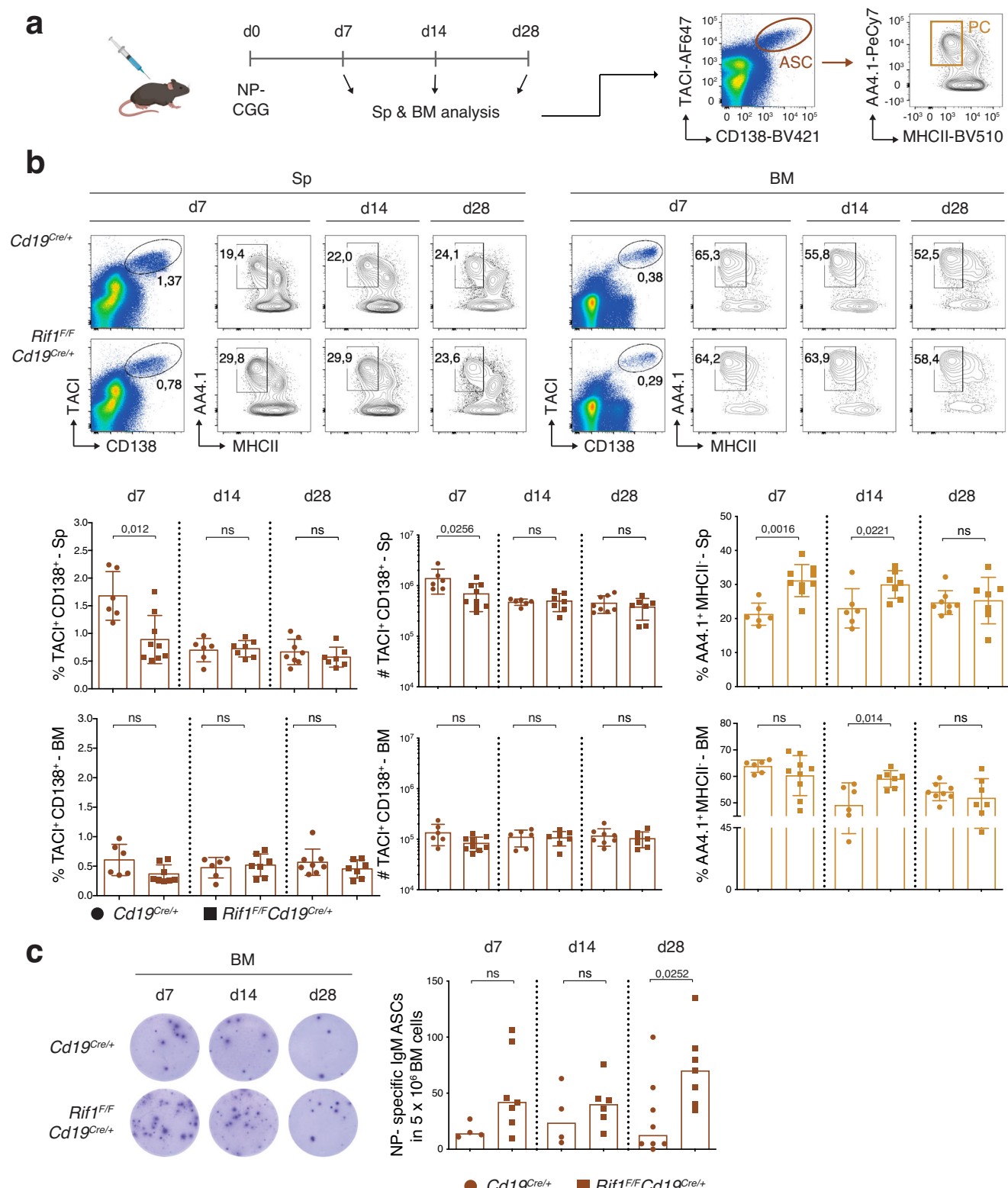

lymphocyte activation, function, and differentiation (Fig. 6c, d, and Supplementary Data 4). Additionally, de novo motif discovery analysis of RIF1 peak sequences revealed consensus binding motifs associated with multiple transcriptional regulators (Fig. 6e and Supplementary Data 5). Altogether, these findings indicate that in activated B cells, RIF1 associates with cis-regulatory elements of genes involved in the modulation of the adaptive immune response.

## RIF1 counteracts the premature repression of BLIMP1 target genes

To further dissect how RIF1 modulates the transcriptional program driving B cell differentiation into ASCs, we next explored the relationship between RIF1 and BLIMP1, the key transcriptional regulator of terminal B cell differentiation[10,34]. To this end, we first assessed the two factors' relative genome occupancy by comparing RIF1 peaks with BLIMP1-bound regions in activated B cells (Fig. 7a, b, and "Methods"). We found

**Fig. 4 | Ablation of RIF1 enhances plasma cell formation after immunization.**
**a** Schematic representation of the NP-CGG immunization protocol and gating strategy employed for the phenotypic analysis of the plasma cell compartment. d: day; NP CGG: 4-hydroxy-3-nitrophenylacetyl hapten conjugated to Chicken Gamma Globulin; Sp: spleen; BM: bone marrow; ASC: antibody secreting cell; PC: plasma cell. **b** Top: Representative flow cytometry plots measuring percentage of antibody secreting cells and plasma cells in spleens and bone marrows of *Cd19*[Cre/+] and *Rif1*[F/F]*Cd19*[Cre/+] mice at the indicated days after immunization. Bottom: Summary graphs for n = nine *Rif1*[F/F]*Cd19*[Cre/+] mice at d7, eight *Cd19*[Cre/+] at d28, seven *Rif1*[F/F]*Cd19*[Cre/+] at d14 and d28, six *Cd19*[Cre/+] at d7 and six *Cd19*[Cre/+] at d14 in at least three independent experiments. **c** Left: Representative ELISpot analysis of NP-specific IgM ASCs in the BM at the indicated times after immunization. Right: Summary graph showing the number of NP-specific IgM ASCs per $5 \times 10^6$ BM cells for n = eight *Cd19*[Cre/+] mice at d28, seven *Rif1*[F/F]*Cd19*[Cre/+] at d7 and d28, six *Rif1*[F/F]*Cd19*[Cre/+] at d14, and four *Cd19*[Cre/+] at d7 and d14 in at least three independent experiments. Significance in panels b and c was calculated with a two-sided Mann−Whitney U test. ns: not significant. Data in panel b are presented as mean values ± SD whereas the graph in panel (**c**) shows the median of each dataset. Source data are provided as a Source Data file. Created in BioRender. Di virgilio, M. (2025) BioRender.com/i30o239.

that 1300 genomic regions (corresponding to 1144 genes) were co-occupied by the two factors (Fig. 7b). These regions were enriched with consensus motifs for BLIMP1 and several members of the interferon regulatory factor (IRF) family (Supplementary Data 6), and exhibited a histone modification profile characteristic of active transcription (Fig. 7c). Sites of AID off-target activity and early replication fragile sites (ERFSs), which represent the genomic hotspots for recurrent DSBs in activated B cells[62,63], accounted for only a fraction of these genomic locations (Fig. 7d). Collectively, these results indicate that RIF1 and BLIMP1 primarily co-occupy regions comprising active genes that are not, for the large part, hotspots for DSB formation in B cells.

Since gene binding is not indicative per se of regulatory activity, we next investigated the link between gene occupancy and transcriptional output. To this end, we first asked whether RIF1 deficiency affects the transcriptional status of BLIMP1 targets, which are defined as BLIMP1-occupied genes that are either up- (repressed, 121 targets) or down- (activated, 93 targets) regulated following its ablation[10]. We found that several BLIMP1-activated genes were significantly up-regulated in *Rif1*[F/F]*Cd19*[Cre/+] cells at 96 h post-activation (Fig. 8a), which is line with the increased differentiation potential of these cultures. More interestingly, BLIMP1-repressed targets were tendentially down-regulated in the absence of RIF1, with the phenotype being evident from earlier time points after activation (Fig. 8a).

Next, we assessed RIF1 occupancy at BLIMP1 targets that are differentially expressed in the absence of RIF1. We found that a considerable portion of BLIMP1 repressed targets that are prematurely down-regulated in the absence of RIF1 was also occupied by RIF1 in wild-type (*Rif1*[FH/FH]) cells at each time point post-activation, which is indicative of a direct RIF1 modulatory function on these genes (Fig. 8b–d, and Supplementary Data 7). This subset of targets comprises several genes whose repression has already been phenotypically and/or functionally linked to late B cell differentiation, including *Nedd4*, *Cd22*, *Ccr7*, *Btg1*, *Sell*, *B3gnt5*, *Bank1*, *Notch2*, and *Id3* (Supplementary Data 7)[10,11,64–73]. In contrast, the subset of BLIMP1 targets that are bound by RIF1 and upregulated in its absence was much more limited and only evident at later time points (Fig. 8b and Supplementary Data 7). Of note, none of the co-regulated repressed targets localized close to ERFSs and only one was in proximity of an AID off-target hotspot (*Klf2*). The differential response of repressed *versus* activated BLIMP1 targets to RIF1 ablation supports a model whereby RIF1 modulates B cell differentiation by acting directly on a subset of repressed BLIMP1 targets rather than by regulating BLIMP1 expression itself. In agreement with this conclusion, CRISPR activation (CRISPRa)-mediated overexpression of RIF1 up to 10 times higher than the physiological levels, does not alter *Prdm1* expression in ex vivo activated B cells (Supplementary Fig. 6).

Altogether, these findings indicate that RIF1 supports the expression of several genes that are physiologically down-regulated by BLIMP1 as part of the transcriptional program promoting B cell differentiation into ASCs.

## Discussion
RIF1 has since long been shown to play a crucial role in humoral immunity because of its ability to protect AID-induced DNA DSBs at

*Igh*, thus ensuring productive class switching events[17–19]. Here, we describe an unexpected modulatory role of RIF1 in regulating the timing of B cell differentiation to PCs that is independent from its DSB end protection function.

Several recent studies have proposed that, besides SHM, BCR diversification via CSR can also influence the differentiation outcome of GC B cells as well as the PC transcriptome[74–79]. Given the strict dependency of CSR on RIF1[17–19], it is tempting to speculate that the skewed differentiation potential of RIF1-deficient B cells might represent an indirect consequence of defective isotype switching. However, the ex vivo recapitulation of the differentiation phenotype in the absence of RIF1 but not of its downstream DNA end protection partner SHLD1 or of AID (Fig. 2), argues against this possibility and in favor of a cell-intrinsic, DNA repair-independent role of RIF1 in fine-tuning PC differentiation kinetics.

Our results support a model where RIF1 serves as a regulator of terminal B cell differentiation through its capacity to counteract the premature repression of BLIMP1 target genes following activation. BLIMP1 is the master regulator of PC differentiation[8,10,80]. Given that upregulation of BLIMP1 expression and appearance of PCs are closely intertwined events, it might appear challenging to unambiguously dissect the direct *versus* indirect aspects of RIF1 modulatory activity. However, the substantial overlap of RIF1 occupancy at BLIMP1 targets that are also differentially regulated in the absence of RIF1 is primarily observed at a subset of BLIMP1 repressed genes (Fig. 8). Furthermore, overexpression of RIF1 in activated B cells does not affect *Prdm1* transcription (Supplementary Fig. 6). These observations suggest that RIF1 exerts its modulatory role on PC differentiation by directly controlling the transcriptional status of these loci rather than by influencing BLIMP1 expression. The dynamic regulation of RIF1 expression in mature B cells, with levels that increase considerably upon activation but decline rapidly thereafter, supports this gene interaction co-regulatory model. We propose that high RIF1 levels prolong an active B cell status to enable efficient *Igh* DSB repair and CSR whereas its down-regulation contributes to a permissive environment for ASC differentiation. Mechanistically, the multiple modalities of RIF1 interaction with BLIMP1 target genes (Fig. 8c) indicate that RIF1 and BLIMP1 might exert their gene co-regulatory function not only by binding to the same DNA regions but also by interacting with distinct cis-regulatory elements. Indeed, although motif enrichment analysis of co-occupied regions identified BLIMP1 binding site as the top-ranked motif (Supplementary Data 6), RIF1 peaks span larger DNA regions and contain consensus motifs for a high number of transcription factors (Fig. 7a and Supplementary Data 5). These observations indicate that RIF1 is recruited to chromatin, and in turn modulates gene expression, possibly in association with different transcriptional regulators. Hence, our study has uncovered the existence of a fine-tuning regulatory mechanism that differentially impinges on the BLIMP1 transcriptional program controlling late B cell differentiation.

Previous reports have highlighted RIF1's role in the differentiation of embryonic stem cells[35,38,39,49]. In this context, RIF1 interacts with multiple Polycomb-group (PcG) proteins, and alters chromatin accessibility and target gene expression[39,48,49]. Additionally, competition between RIF1 and KAP1 (reinforced by the long non-coding RNA

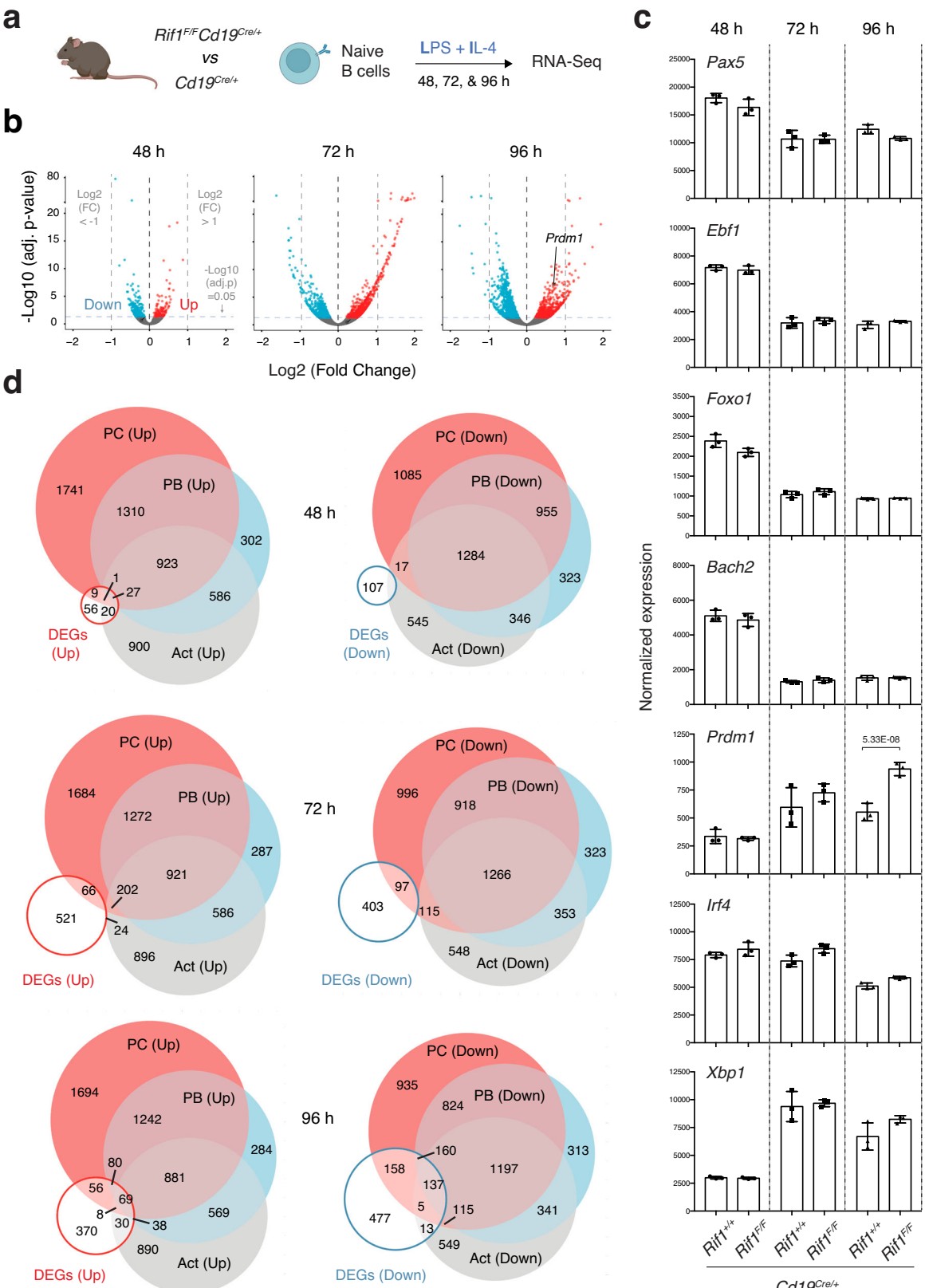

(lncRNA) *Tsix*) for interaction with *Xist* promoters establishes the asymmetric X-chromosome state at the onset of embryonic stem cell (ESC) differentiation[48]. The interplay between RIF1, KAP1 and the *cis*-acting lncRNAs *Xist* and *Tsix* not only exemplifies RIF1's role as a direct and positive regulator of transcription, but it also highlights its contribution to chromatin state modulation during a key ESC transition.

Therefore, our findings and proposed model in activated B cells support the existence of widespread mechanisms for RIF1-dependent transcriptional modulation across cell types and differentiation states.

The multifaceted role of RIF1 underscores the evolutionary ingenuity in repurposing a single protein for diverse, yet equally critical, biological functions. RIF1 moonlighting in mature B cells has important

**Fig. 5 | RIF1 deficiency skews the transcriptional profile of activated B cells towards ASCs. a** Schematic representation of gene expression analysis in naïve B cells isolated from *Cd19*[Cre/+] and *Rif1*[F/F]*Cd19*[Cre/+] mouse spleens and stimulated ex vivo with LI for 48, 72, and 96 h (n = three). For the 48 h time point, the analysis was performed also on LBT-stimulated cultures. **b** Volcano plots displaying differentially expressed genes between control and RIF1-deficient splenocytes. The blue and red dots represent transcripts down- and up-regulated (adjusted *p* value <= 0.05, blue horizontal dotted line), respectively, in *Rif1*[F/F]*Cd19*[Cre/+] cells. The gray vertical dotted lines mark the log2 FC values of -1 and 1. **c** Expression of *Pax5*, *Ebf1*, *Foxo1*, *Bach2*, *Irf4*, *Xbp1*, and *Prdm1* as determined by the RNA-Seq. Data were normalized by DESeq2 and are presented as mean values ± SD. The adjusted *p* value of the only significant difference between samples is indicated. **d** Venn diagrams depicting the overlaps between genes up- (left) and down- (right) regulated in RIF1-deficient B cells at the indicated times post-activation and the corresponding up- and down-regulated (over naïve B cells) categories in the activated (Act) B cell, PB, and PC transcriptional signatures (Minnich et al. 2016). Significance in panels b and c was calculated using a two-sided Wald test, with correction for multiple testing applied through the Benjamini-Hochberg procedure. Source data are provided as a Source Data file. Created in BioRender. Di virgilio, M. (2025) BioRender.com/ o81f217.

implications for the pathological consequences of deregulated plasma cell generation. Whilst producing a diverse Ig-switched repertoire enables effective antibody-mediated responses, tight regulation and fine tuning of late B cell differentiation is essential to counteract the development of autoimmunity and PC-derived malignancies[81–83]. Hence, by enabling *Igh* diversification via CSR while exerting a modulatory function on PC differentiation, RIF1 integrates key requirements for the establishment of protective humoral immunity.

## Methods

### Mice and derived primary cell cultures

*Rif1*[FH/FH] [62], *Cd19*[Cre52], *Rif1*[F/F]*Cd19*[Cre/+18], *Shld1*[-/-45] and *Aicda*[-/-55] mice were previously described and maintained on a C57BL/6 background. *Rosa26*[dCas9-Suntag/+] mice were generated by breeding Rosa26-LSL-dCas9 mice, purchased from the Jacksons laboratory (RRID: MMRRC_043926-JAX), with BALB/c-Tg(CMV-cre)1Cgn/J mice. Mice were kept in a specific pathogen-free (SPF) barrier facility and all experiments were performed in compliance with the European Union (EU) directive 2010/63/EU, and in agreement with Landesamt für Gesundheit und Soziales directives (LAGeSo, Berlin, Germany). Mice of both genders were used for the experiments.

Resting B lymphocytes were isolated from mouse spleens using anti-CD43 MicroBeads (Miltenyi Biotec), and grown in RPMI 1640 medium (Life Technologies) supplemented with 10% fetal bovine serum (FBS, Sigma-Aldrich), 10 mM HEPES (Life Technologies), 1 mM Sodium Pyruvate (Life Technologies), 1X Antibiotic Antimycotic (Life Technologies), 2 mM L-Glutamine (Life Technologies), and 1X 2-Mercaptoethanol (Life Technologies) at 37 °C and 5% CO2 levels. Naïve B cells were activated by addition of 5-25 µg/ml LPS (Sigma-Aldrich) and 5 ng/ml of mouse recombinant IL-4 (Sigma-Aldrich) (L-I), or 5 µg/ml LPS, 10 ng/ml BAFF (PeproTech) and 2 ng/ml TGFβ (L-B-T), or 5 µg/ml LPS only (L).

### CRISPRa

The retroviral pMSCV-gRNA-scfv-VP64-eGFP vector was built by combining the scfv fragment obtained from the phr-scfvGCN4-sfGFP-GB1-NLS-dWPRE vector (addgene #60906) and the VP64 fragment from the PB-TRE-dCas9-VPR vector (addgene #63800) into the MSCV_hU6_CcdB_PGK_Puro_T2A_BFP vector[84] and BFP was replaced with eGFP. gRNAs against the promoter regions of *Rif1* and *Prdm1* as well as gRNAs targeting regions not present in the mouse genome (*gRandom*) (Supplementary Table 1 within Supplementary file pdf) were cloned into pMSCV-gRNA-scfv-VP64 using the two BbsI cloning sites. Individual gRNA-containing constructs were transfected together with pCL-Eco into the HEK293T derivative cell line BOSC23 using FuGENE HD Transfection Reagent (Promega). Naïve B cells isolated from *Rosa26*[dCas9-Suntag] mice (ubiquitously expressing dCas9-P2A-BFP) were activated with 25 µg/ml LPS, 5 ng/ml IL-4, and 0.5 µg/ml anti-CD180 (RP/14) (BD Biosciences). 16 h after isolation, primary splenocyte cultures were transduced with the viral supernatant from the transfected BOSC23 cells and samples were sorted for GFP+ cells 72 h later.

Analysis of *Prdm1* and *Rif1* transcript levels was performed as it follows. Total RNA was extracted from sorted GFP+ cells using TRIzol

(Invitrogen) and retro-transcribed with iScript™ cDNA synthesis kit (BioRad) according to the manufacturer's instructions. Genomic DNA was removed with RapidOut DNA Removal Kit (Thermo Fisher Scientific). cDNA was amplified using Luna Universal qPCR Mastermix (NEB) in a StepOnePlus Real-Time PCR System (Applied Biosystems) and values were normalized against the expression of the *Ubc* gene. Primers used for qPCR are listed in Supplementary Table 1 within Supplementary file pdf.

### Western blot

WB analysis of protein levels was performed on whole-cell lysates prepared by lysis in radioimmunoprecipitation assay buffer (Sigma-Aldrich) or 0.5% NP-40 buffer (20 mM Tris-HCl, 150 mM NaCl, 0.5% NP-40 and 1.5 mM MgCl₂) supplemented with Benzonase (Sigma-Aldrich), Complete EDTA free proteinase inhibitors (Roche) and Pierce Phosphatase Inhibitor Mini Tablets (Thermo Fisher Scientific). The antibodies used for WB analysis are anti-RIF1[18], anti-BLIMP1 (Novus NB600-235, 1:1000), and anti-β-actin (Sigma Aldrich A5441, 1:100000). Full scanned blots are provided in the Source Data file and Supplementary file pdf.

### RNA-Seq

For each RNA-Seq dataset, the analysis was performed on three mice per genotype. Splenocytes were cultured in LPS and IL-4 (LI), LPS, BAFF and TGFβ (LBT), or LPS (L), and cells were collected at the indicated time points by centrifugation. RNA was extracted with TRIzol (Invitrogen) according to manufacturer's instructions, and ribosomal RNA was depleted using Ribo-Zero Gold rRNA Removal Kit (Illumina) for all datasets except for the RNA-Seq analysis in naïve versus LI/LBT/L-activated splenocytes from WT mice (Fig. 1 and Supplementary Fig. 1), for which RNase H (Epicenter) treatment was used. Libraries were prepared with TruSeq Stranded Total RNA Library Prep Kit Gold (Illumina), and run in one lane on a flow cell of NovaSeq 6000 SP (Illumina).

### ChIP-Seq

ChIP-Seq for RIF1 in LPS and IL-4-stimulated splenocyte cultures was previously described[34]. H3K4me3 ChIP-Seq was performed in splenocytes activated with LPS and IL-4 for 72 h, and employed anti-H3K4me3 antibody (abcam, ab8580, 10µg of antibody per 10 million cells) for the ChIP part of the previously described protocol[85]. For H3K27me3 ChIP-Seq, we used H3K27me3 antibody (Cell Signaling, C36B11, 10µg of antibody per 10 million cells), and 2.5% human shared chromatin was spiked into all samples as an internal reference for normalization[86].

### ChIP-qPCR

25 µg of chromatin was used for immunoprecipitation as previously described[87]. For bead washing, elution and reverse cross-linking SimpleChIP® (Cell Signaling, 9003) kit was used according to the manufacturer's instructions. For ChIP-qPCR, 1 µl of purified 1% input and ChIP DNA samples was used for amplification of the selected RIF1-occupied as well as RIF1 and BLIMP1 co-occupied peaks. Regions without detected peaks were used as a negative control. DNA was amplified using Luna Universal qPCR Mastermix (NEB) in a StepOnePlus Real-Time PCR System (Applied Biosystems). Primers are listed in Supplementary Table 1 within Supplementary file pdf.

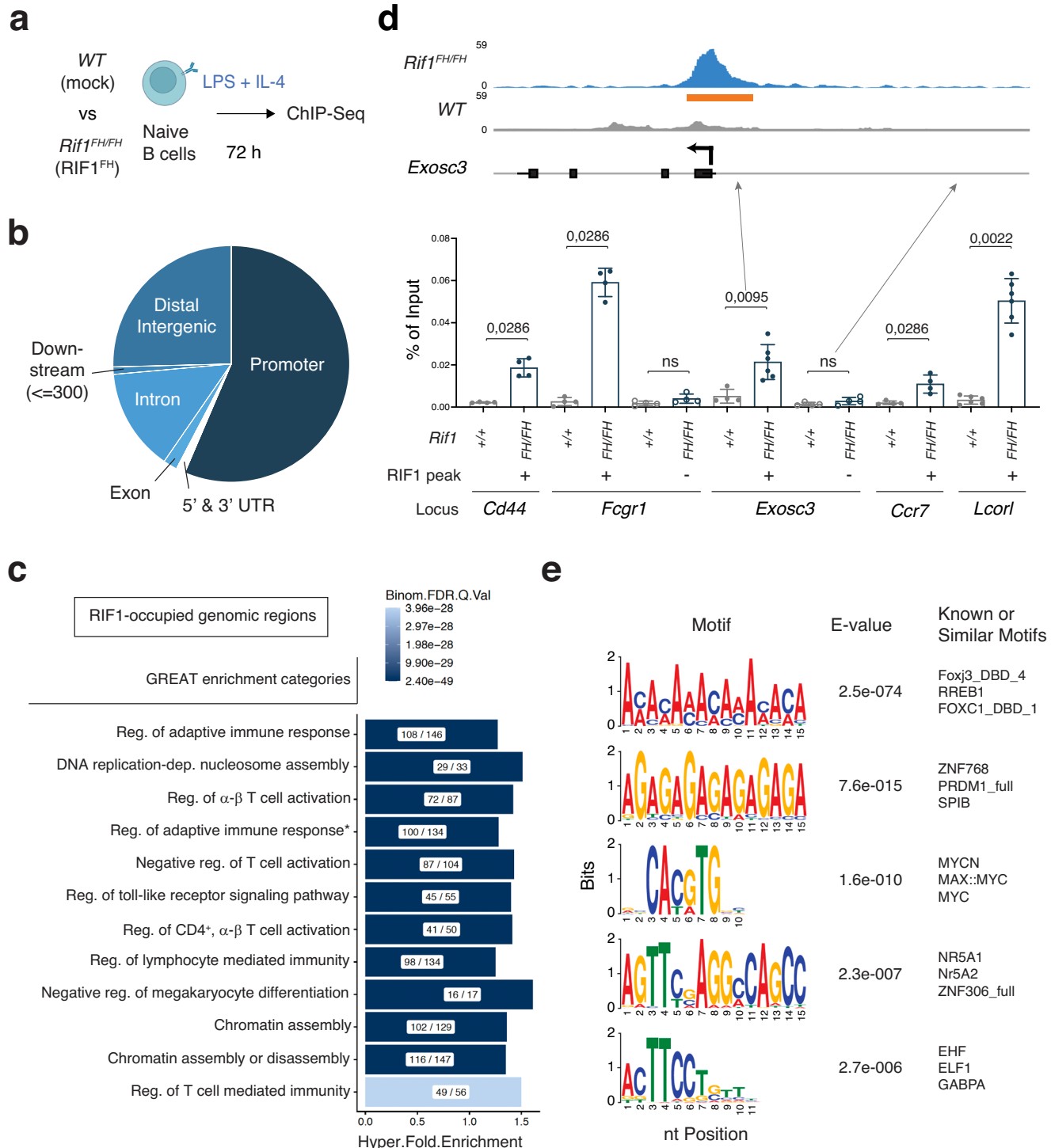

**Fig. 6 | RIF1-occupied genomic regions in activated B cells comprise genes involved in lymphocyte activation and differentiation. a** Schematic representation of the ChIP-Seq analysis in mature B cells isolated from spleens of *Rif1^FH/FH* mice and stimulated ex vivo with LPS and IL-4 (LI). **b** Genomic distribution of RIF1-occupied annotated regions. **c** Gene ontology enrichment analysis as determined by GREAT for the genomic regions bound by RIF1 in LI-stimulated B cells. The complete name of the category marked with " \*" is "Regulation of adaptive immune response based on somatic recombination of immune receptors built from immunoglobulin superfamily domain". The X / Y ratio within each bar indicates the number of genes occupied by RIF1 (X) out of the total number of genes in the category (Y). Reg: regulation; FDR: false discovery rate. **d** Graph depicting RIF1 occupancy as determined by ChIP-qPCR at few loci selected from the first category of the GREAT analysis in panel (**c**) (n = six for *Exosc3* (positive region) and *Lcorl*, and

four for the remaining loci). The "+" and "-" denote presence and absence, respectively, of RIF1 peaks as seen in the RIF1 ChIP-Seq tracks. The graph summarizes data from at least four mice per genotype in three independent experiments. The inset over the graph shows the RIF1 ChIP-Seq tracks from *WT* and *Rif1^FH/FH* B cells at a representative locus (*Exosc3*). The orange box delineates the analysed RIF1 peak as determined by RIF1 ChIP-Seq. **e** The five highest-ranked RIF1 binding motifs identified using the de novo motif discovery tool MEME-ChIP. For each motif, the logo, significance (E value), and the three most similar motifs are provided. Significance in panel d was calculated with a two-sided Mann–Whitney U test, with data presented as mean values ± SD. ns: not significant. Source data are provided as a Source Data file. Created in BioRender. Di virgilio, M. (2025) BioRender.com/g40z397.

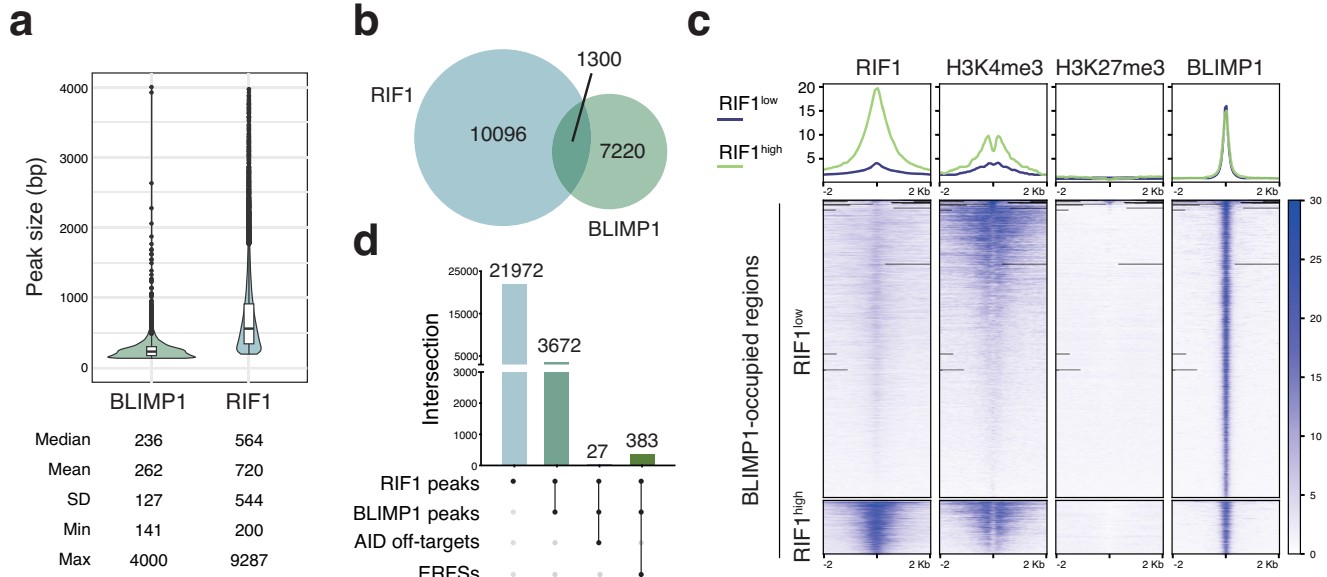

**Fig. 7 | RIF1 and BLIMP1 co-occupy transcriptionally active chromatin. a** Violin and box plot showing the median and quantile distribution of BLIMP1 and RIF1 peak size. SD: standard deviation; Min: minimum peak size; Max: maximum peak size. **b** Venn diagram depicting the overlap between RIF1- and BLIMP1-bound genomic regions in activated B cells. **c** Line plots (top) and heatmaps (bottom) depicting the comparative genome distribution of RIF1, H3K4me3, and H3K27me3 in reference to BLIMP1-occupied regions. The line plots illustrate the mean normalized signal distribution of the indicated proteins, whereas the heatmaps visualize their signal strength at each identified peak. The x-axis in both graph types represents the genomic region relative to the peak center. N = two biological replicates for H3K27me3 ChIP-seq, two biological replicates per genotype for RIF1 ChIP-seq, and one biological replicate for H3K4me3 ChIP-seq. **d** UpSet plot depicting the intersection between RIF1 peaks, BLIMP1 peaks, AID off-targets, and early replication fragile sites (ERFSs). The dot matrixes at the bottom of each graph show the intersection relationships among the data sets, with the number of common elements in the intersecting sets indicated above each bar.

The percent input method was used to calculate the enrichment of the selected regions. Briefly, a dilution factor of 100 (6.644 cycles) was subtracted from the Ct value of the 1% input samples, and the adjusted input values were used to normalize the values of their respective ChIP samples.

### B cell proliferation, development and differentiation analyses

For analysis of CSR in ex vivo cultures, cell suspensions were stained with anti-IgG1-APC (BD Pharmigen, 560089, clone A85-1, 1:200), anti-IgG3-Biotin (BD Pharmigen, 553401, clone R40-82, 1:400) and Streptavidin-APC (BioLegend, 405207, 1:600), or anti-IgG2b-PE (BioLegend, 406708, clone RMG2b-1,1:400). For analysis of plasmablast differentiation ex vivo, isolated naïve splenic B cells were cultured at a density of 0,5 ×10⁶ cells/ml in the presence of 25 µg/ml LPS and 5 ng/ml IL-4 and stained with anti-CD138 (CD138 BioLegend, 142503 or 142508, clone 281-2, 1:200). B cell proliferation was assessed by CellTrace Violet (Thermofisher) dilution according to manufacturer's instructions. For analysis of plasma cell-like differentiation ex vivo, the induced GC B (iGB) culture system was used as described before[51,53]. Briefly, 40LB feeder cells (Balb/c 3T3 cell line expressing exogenous CD40-ligand (CD40L) and B-cell activating factor (BAFF)) were irradiated with 80 Gy and co-cultured with primary splenic B cells for 4 days in high glucose DMEM medium (Gibco) supplemented with 10% FBS, 10 mM HEPES, 1 mM Sodium Pyruvate, 1X Pen/Strep (Life Technologies), 1x MEM Non-Essential Amino Acids (Life Technologies), 2 mM L-Glutamine, 1X 2-Mercaptoethanol and 1 ng/ml IL-4 at 37 °C and 5% CO₂ levels. On day 4, cells were harvested, washed one time with PBS and re-plated in a newly irradiated 40LB feeder layer in the presence of 10 ng/ml IL-21 (PeproTech) for the next 2, 3 or 4 days. For assessing CSR and plasma cell-like differentiation at the respective culture timepoints, 1 × 10⁶ B cells were washed one time with FACS buffer (PBS supplemented with 1% FCS and 1 mM EDTA), blocked with TruStain fcX (BioLegend) for 10 min at room temperature and stained with anti-IgM (eBioscience, 25-5790-82, clone II/41, 1:200), anti-IgG1

and anti-CD138. 1 µg/ml of propidium iodide (PI) was used for live/dead cell staining.

For analysis of germinal centers and plasma cell compartments in vivo, 8-14 week-old mice were sacrificed to isolate the spleen, tibia, and Payer's patches (PP). Single cell suspensions from spleen and tibia were incubated for 2 min with ACK buffer (Gibco) for red blood cell lysis. For surface staining, 5-7 × 10⁶ cells were first blocked with TruStain fcX for 10 min at 4 °C and then stained for 20 min at 4 °C. The antibodies employed for the staining were detecting: CD138, CD267/ TACI (BD Pharmigen, 558453, clone 8F10, 1:200), MHC-II (BioLegend, 107635, clone M5/114.15.2, 1:200) and CD93/AA4.1 (BioLegend, 136506, clone AA4.1, 1:200) for the plasma cell compartment; CD19 (BioLegend, 115539 or 115541, clone 6D5, 1:200), B220 (BioLegend, 103245, clone RA3-6B2, 1:200), CD86 (BioLegend, 105032, clone GL-1, 1:200), CD184/ CXCR4 (CXCR4-biotin, BD Pharmigen, 551968 1:200 and streptavidin BioLegend 405207 or 405229, 1:600), CD95/FAS (BD Pharmigen, 557653, clone Jo2, 1:200), IgA (eBioscience, 12-4204-82, clone mA-6E1, 1:200) and CD38 (eBioscience, 56-0381-82, clone 90, 1:200) for PP analysis; and CD19, B220, CD38, CD95/FAS and IgG1 (BD Biosciences, 740121, clone A85-1, 1:200) for splenic GC analysis. Cells were resuspended in FACS buffer containing PI and analyzed. Immunization was performed by intraperitoneal injection of 100 µg of NP-CGG (Biosearch Technologies; ratio 10–20) precipitated in alum (Sigma).

All samples were acquired on a LSRFortessa cell analyzer (BD Biosciences).

For analysis of conditional *Rif1* allele deletion, single cell suspensions from tibia and spleen were blocked with TruStain fcX for 10 min at 4 °C, stained with anti-B220 (Biolegend) and anti-IgM (eBioscience) for 20 min at 4 °C, and bulk-sorted using a BD FACSAria I flow cytometer. Genomic DNA from purified bone marrow Pre-Pro B cells (BM B220⁺IgM⁻), bone marrow immature B cells (BM B220⁺IgM⁺), splenic non-B cells (Sp B220⁻IgM⁻) and splenic B cells (Sp B220⁺IgM⁺) was isolated using the Quick DNA extraction buffer (Lucigen). WT, floxed *Rif1*, and the deleted null alleles were detected by PCR using the primers

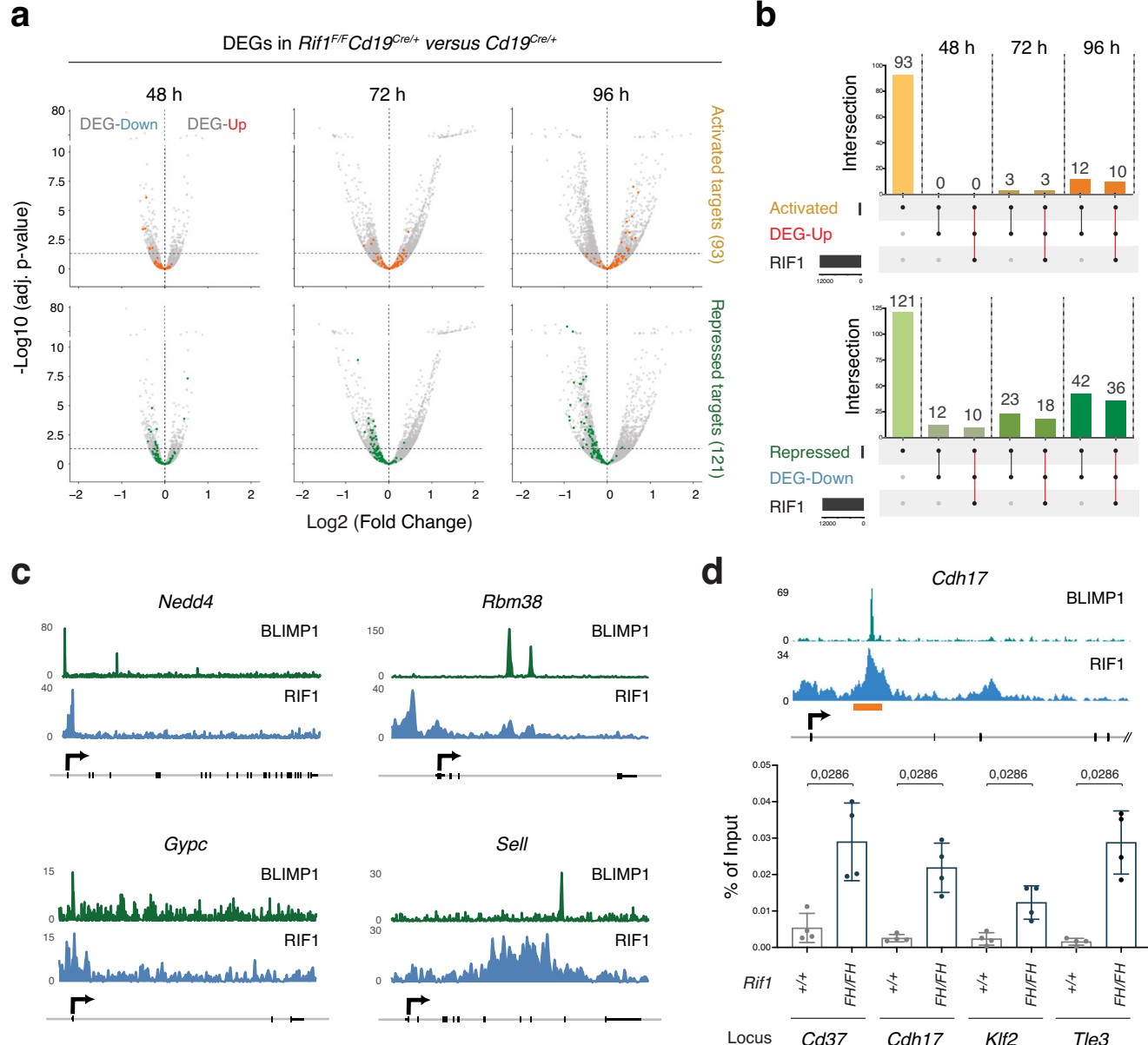

**Fig. 8 | RIF1 counteracts the premature repression of BLIMP1 target genes.**
**a** Volcano plots displaying BLIMP1-activated (in orange, top) and -repressed (in green, bottom) target genes among the transcripts (in gray both up- and down-regulated) identified in the *Cd19^Cre/+^* versus *Rif1^F/F^Cd19^Cre/+^* transcriptome analysis shown in Fig. 5. The horizontal dotted line in the plots denotes the adjusted *p* value of 0.05. **b** UpSet plots depicting the intersection between activated BLIMP1 targets (Activated), genes differentially upregulated in *Rif1^F/F^Cd19^Cre/+^* B cells (DEG-Up) and RIF1-occupied regions (top), and between repressed BLIMP1 targets (Repressed), genes differentially down-regulated in *Rif1^F/F^Cd19^Cre/+^* B cells (DEG Down) and RIF1-occupied regions (bottom). The dot matrixes at the bottom of each graph show the intersection relationships among the data sets, with the number of common elements in the intersecting sets indicated above each bar. The red lines in the matrixes highlight the group of genes occupied and regulated by both BLIMP1 and RIF1. **c** ChIP-Seq tracks for BLIMP1 (Minnich et al., 2016) and RIF1 at representative co-occupied BLIMP1 repressed targets from panel (**b**). **d** Graph depicting RIF1 binding to regions overlapping with BLIMP1 peaks at selected BLIMP1 repressed targets as determined by ChIP-qPCR. The graph summarizes data from four mice per genotype in two independent experiments. The inset over the graph shows BLIMP1 and RIF1 tracks at a representative locus (*Cdh17*). The orange box delineates the analysed RIF1 peak as determined by RIF1 ChIP-Seq. Significance in panel (**a**) was calculated using two-sided Wald test, with adjustments for multiple comparisons made through the Benjamini-Hochberg procedure. Significance in panel d was calculated with a two-sided Mann–Whitney U test, with data presented as mean values ± SD. Source data are provided as a Source Data file.

listed in Supplementary Table 1 within Supplementary file pdf. PCR band intensity was quantified using Image J.

**ELISpot assay**
To detect NP-specific plasma cells, MultiScreen_HTS_-IP plates (Merck) were activated with 35% ethanol in PBS, washed three times with PBS, and finally coated with 2 μg/mL NP-BSA (loading ratio >20, BioCat) at 4 °C overnight. Next day, plates were washed twice with PBS and blocked with ELISpot medium (RPMI 1640 supplemented with 10% FBS

and 1% pen/strep) for 2 h at 37 °C. 5 × 10^6 total spleen and BM cells (as well as serial fivefold dilutions) were added to the NP-BSA coated plates and cultured overnight in ELISpot medium. The following day, wells were washed six times with PBS supplemented with 0.1 % Tween20 and incubated for 2 h with 1 μg/mL biotinylated goat α-mouse IgM (Southern Biotech, 1020-08) or IgG1 (Southern Biotech, 1071-08) at 37 °C. Spot visualization was performed by incubation with 0.3 U/ml streptavidin-AP conjugate (Roche, 11089161001) for 30 min at room temperature followed by three washes with running distilled

water, equilibration in AP buffer (100 mM Tris-HCl, pH 9.0, 150 mM NaCl, and 1 mM MgCl$_2$), and development using NBT/BCIP substrate mix (Promega) diluted in AP buffer. Plates were scanned and spots were counted using the ImmunoSpot® Series 6 Alfa Analyzer and ImmunoCapture™ Image Acquisition as well as ImmunoSpot® Analysis software (C.T.L.).

## Availability of biological materials

Mouse models and plasmids are available upon request to Michela.divirgilio@mdc-berlin.de.

## Data analysis and softwares

*RNA-Seq*. All RNA-Seq datasets, whether specifically generated for this study or obtained from public databases, have been analyzed/re-analyzed using the same pipeline implemented for this project. The RNA-seq data was analyzed using the Galaxy platform[88]. HTSeq[89] mapped the reads to the GRCm38 assembly for mouse and GENCODE-M25 (www.gencodegenes.org) was used to count the transcript abundance. Differential expression analysis was done via the DESeq2[90] package for R, which uses the Wald test for significance. To define the transcriptional signatures of activated B cells, PBs, and PCs, the analysis was performed using the naïve B cell RNA-Seq dataset from the same study[10] for baseline comparison. Expression of differentially expressed genes were obtained from Immgen database[50] and visualize by R and ggolot2 packages (https://ggplot2.tidyverse.org/) after scaling. For the comparative assessment of *Cd19^{Cre/+}* and *Rif1^{F/F}Cd19^{Cre/+}* B cell transcriptomes at 48 h after activation, RNA-seq datasets from L-I and L-B-T splenocytes cultures were merged. The increase in sample size (from 3 to 6 repeats) for both groups enhanced statistical power to detect minor gene expression differences. The multiple treatments are used to discern genotype effects, thus ensuring that the observed differences are due to the absence of RIF1.

*ChIP-Seq*. ChIP-Seq and BLIMP1 Bio-ID datasets have been analyzed/re-analyzed using the same pipeline implemented for this project. FASTQ files were aligned against mouse genome (mm10) using BWA aligner[91]. Processing and peak-calling of ChIP-Seq data were performed with MACS2[92]. Peak annotation was done using R and ChIPseeker package[93]. Functional assessment of genomic regions enrichment were performed by GREAT[61]. Motif discovery on RIF1-bound regions was performed using MEME-ChIP (v5.5.7) from the MEME Suite[94], using default parameters with the exception of the expected motif site distribution, which was set to "Any number of repetitions." The input sequences were analyzed for enriched motifs, and additional motif-based analyses were carried out using the integrated tools within the MEME Suite. For motif enrichment analysis of regions with overlapping RIF1 and BLIMP1 peaks (>= 100 bp), the SEA tool[95] from the MEME Suite (v5.5.7) was employed using default parameters. The input motifs were compared against a pre-defined database to identify statistically significant motif enrichment in the provided sequences. Genomic data from AID off-targets[62], ERFSs[63], RIF1 and BLIMP1 peaks was analyzed to identify overlapping regions as it follows[96]. BED files were processed into GRanges objects using the GenomicRanges package in R, with AID off-targets and ERFSs lifted from mm9 to mm10. Overlaps between AID off-targets, ERFSs, RIF1 and BLIMP1 peaks were identified using the findOverlaps function. The genomic regions overlapping across AID off-targets, RIF1 and BLIMP1 peaks and across ERFSs, RIF1 and BLIMP1 peaks were uploaded to the GREAT tool for functional annotation and nearest gene identification. These results were compared with known BLIMP1 target genes[10]. All analyses were conducted using R, employing the GenomicRanges and ggVennDiagram packages. For the comparative genomic distribution analysis of RIF1, H3K4me3, and H3K27me3 in reference to BLIMP1-occupied regions, the aligned reads were converted into BigWig format. The signals were subsequently transformed into a matrix via the ComputeMatrix software, and visualized as a heatmap[97].

## Statistical analysis

Statistical details of experiments can be found in the figure legends.

**Hypergeometric analysis.** Gene expression data were sourced from experiments from this study and published datasets[10] (GSE71698) and compiled into a single CSV in parallel to a universe of genes (list of all known genes in mouse genome, mm10). A custom R function was developed to conduct overrepresentation analysis via hypergeometric testing. This function computes the likelihood of observing the intersection of genes between two lists, considering the total gene count in the universe and adjusting for the sizes of the lists involved. Pairwise comparisons between all unique pairs of gene lists were conducted by the hypergeometric test, which was implemented using the phyper function from R's base stats package, with the test's directionality set to identify overrepresented genes. The resulting p-values were adjusted for multiple comparisons using the Bonferroni correction method to control for the family-wise error rate. For a comprehensive overview of the pairwise comparison results, we reported the Bonferroni-adjusted p-values as -log10 values. This transformation enhances the interpretability of the results, with higher values indicating more significant overlaps between gene list pairs.

**Source codes for data analysis.** All codes used to generate graphs and to compare different datasets can be found on our GitHub page (https://github.com/arahjou/RIF1_paper.git)[98].

## Reporting summary

Further information on research design is available in the Nature Portfolio Reporting Summary linked to this article.

## Data availability

All RNA-Seq datasets reported in this study, and H3K4me3 and H3K27me3 ChIP-Seq data have been deposited in the GEO repository under accession number GSE237560. The transcriptional signatures of activated B cells, PBs, and PCs have been defined using the corresponding (and naïve B cells for baseline comparison) RNA-Seq datasets from ([10], GSE71698, [https://www.ncbi.nlm.nih.gov/geo/query/acc.cgi]). RIF1 ChIP-Seq was previously reported ([34], GSE228880, [https://www.ncbi.nlm.nih.gov/geo/query/acc.cgi]). The BLIMP1-bound regions in activated B cells were defined based on the BLIMP1 Bio-ID dataset from ([10], GSE71698). The lists of BLIMP1-activated and -repressed targets have been previously described[10]. Source data are provided with this paper.

## Code availability

All codes used to generate graphs and to compare different datasets can be found on https://doi.org/10.5281/zenodo.14030238.

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

## Acknowledgements

We thank all members of the Di Virgilio lab for discussion; L. Keller and T. Rüster (Di Virgilio lab, MDC, Berlin) for genotyping & cloning; W. Winkler and C. Farre i Diaz (Janz/Mathas and K. Rajewsky labs, MDC) for in vivo and ELISpot protocols; M. Serresi (MDC) for her technical insights on CRISPRa; and D. Pasini (IEO, Milan) and G. Gargiulo (MDC) for their

valuable feedback. Figure schematics were created using images from BioRender. The project was funded by the Helmholtz-Gemeinschaft Zukunftsthema "Immunology and Inflammation" ZT-0027 (to M.D.V.) and the Initiative and Networking Fund W2/W3 Program of the Helmholtz Association (to M.D.V.).

## Author contributions

E.K. and A.R. conceived the project idea, designed and performed the majority of experiments; M.B.L., S.E., T.S and V.D.B performed experiments; A.R. analyzed all high-throughput sequencing data; R.A. contributed to the sequencing data analysis; R.P. engaged in active discussions on the study; M.D.V. secured the funding for the project, supervised all aspects of the study, and wrote the manuscript; E.K. and A.R. reviewed and edited the manuscript.

## Funding

## Competing interests

The authors declare no competing interests.
