## [Transparent Peer Review file · Nature Communications]

RIF1 integrates DNA repair and transcriptional requirements during the establishment of humoral immune responses

Corresponding Author: Professor Michela Di Virgilio

Version 0:

Reviewer comments:

Reviewer #1

(Remarks to the Author)

RahjoueiDiVirgilio et al.

A series of programmed differentiation steps driven by various transcription factors (TFs) orchestrates the development of common lymphoid progenitor (CLPs) to antibody secreting B cells (plasmablasts (PB) or plasma cells (PC)). Various transcription factors and other transcriptional regulatory events drive each of the developmental steps, during early B cell development in the bone marrow, in the germinal center and finally during ASC differentiation in the periphery. RIF1 protein has been previously shown to be important for prevention of resection of DNA double strand breaks and promote NHEJ during class switch recombination, a DNA recombination step that defines the constant region of antibodies expressed by ASCs. In this manuscript, the authors aim to establish that in addition to DNA break end repair/resection suppression activity, RIF1 also functions directly as a transcription regulatory factor by binding to BLIMP1 sites, a TF that promotes APC differentiation, and thereby regulates the kinetics of APC development.

The authors have nicely demonstrated that B cells increase RIF1 expression during late B cell differentiation (Fig. 1). RIF1 depletion leads to increased kinetics of APC differentiation *in vivo* and *ex vivo* (Fig. 2) without affecting germinal center dynamics (Fig. 3). *In vivo*, immunization of mice deficient of RIF1 leads to increased level of plasma B cell formation (Fig. 4). Finally, a series of experiments evaluating genomic occupancy of RIF1 using ChIP experiments indicates that RIF1 overlaps BLIMP binding sites and therefore antagonizes BLIMP function to enhance ASC/PC differentiation (Fig. 6 and 7).

Overall, I found the data quite important, compelling and the manuscript very well written. Since it is assumed that RIF1 functions in regulating DNA resection, its functions as a transcription regulator is a surprise and can be considered as a novel discovery. Thus, I am supportive of publication of this study. Below I list a few questions that the authors could consider addressing during revisions.

1. What is the effect of RIF1 over expression on *in vitro* cultured GC B cells in terms of APC differentiation. If the experiment in Fig. 2d and 2e is performed with RIF1 over expressor cells, will that attenuate CD138+ B cell levels.
2. Will RIF1F/F B cells accumulate genetic alterations or gene expression changes seen enriched in myeloma B cells.
3. Is there a difference in kinetics of PC differentiation of switched and unswitched cells obtained from B cells that are depleted of RIF1 (RIF1F/F. CD19Cre).
4. In Fig. 7e, does the occupancy of BLIMP1 increase at target genes following RIF1 depletion. How are DNA breaks induced RIF1 peaks separated from those peaks recruited for APC differentiation (BLIMP1 antagonistic peaks). Is that possible to do.
5. Overall, the manuscript is very nicely written. The discussion could cover a few more points. (1) At a molecular and biophysical level how are BLIMP1 and RIF1 competing. The peaks shown in Fig 7 are overlapping partially but not completely. Is there a specific RIF1 binding motif that overlaps with BLIMP1 binding motif identified from the RIF1/BLIMP1 peak calling? Is there another explanation. (2) The aspects of AID mutations on single strand DNA to generate DNA DSBs, how that occurs and how RIF1 controls DNA repair at AID induce DSBs could be covered better. The mechanism of AID, RIF1 and CSR is not discussed properly for a complete understanding of RIF1 function in CSR versus APC differentiation. (3) Finally, the authors mention that RIF1 can alter chromatin accessibility and target gene expression via its interaction with

Polycomb group proteins. Chromatin accessibility is dependent upon basal noncoding RNA expression and on various epigenetic marks, among other factors. Outlining RIF1 collaboration with ncRNA biology and/or epigenetic marks, if possible, will make the discussion interesting (and substantially different than what had already been stated in the introduction and results sections).

Reviewer #2

(Remarks to the Author)

The manuscript by Di Virgilio examines the role of RIF1 in B cell biology. This factor is known to be involved in DNA end protection to facilitate NHEJ. But other literature suggests that it has other functions, including activities in modulating transcriptional networks. In this report, the authors show that RIF1's expression is increased in stimulated B cells. They further show that plasma cells (CD138+) are increased by ~2-fold in RIF KO without an effect on GC dynamics, with the exception of showing the expected effect on isotype switching. The authors further show that RIF KO B cells have about a 2-fold increase in Blimp1 (Prdm1) expression, and that RIF binds to genes involved in the adaptive immune response, and overlaps at many loci also bound by BLIMP1. The authors suggest that RIF1 binds to targets and inhibits BLIMP1 action there, although this last section was not written clearly, so it is hard to interpret the data.

Overall, this is an interesting study, but the overall effects observed in the RIF1 KO were not striking. The manuscript would benefit from a more detailed analysis of RIF1 binding to specific genetic regions (and to the Prdm1 gene, itself) and how this binding modulates BLIMP1 function, as well as BLIMP1 expression since Blimp1 expression is increased in the RIF KOs.

Additional comments

1. The flow cytometry analysis in Fig 4b does not show much of an increase in PCs as a result of RIF1-deletion, as suggested by the authors. In addition, the absolute numbers of cells needs to be reported here and in previous figures (including supplemental figures) as the total cellularity may have been affected in the RIF1 KOs that may give the impression that PC formation is increased in the absence of RIF1.
2. Line 119: Not clear why the authors stated this. To show this effect on increased PB and how it is not related to end protection, the authors need to do double KOs with RIF1 and AID, for example.
3. The RNAseq experiments shown in Fig 5 suggest that many genes are differentially regulated in the KO B cells, but in the text they suggest only a few genes (line 205). What is the discrepancy here? Also, the authors checked for expression of B cell identity genes (Fig 5c): were these selected based on the RNAseq? There was not difference in expression in most genes examined which may or may not disagree with the RNAseq data.
4. The CHIPseq data needs validation, at a minimum at a few loci.

Reviewer #3

(Remarks to the Author)

The establishment of protective immune responses hinges upon the robust capability of mature B cells to secrete a diverse array of antigen-specific antibodies, each endowed with distinct effector functions. At the heart of this process lies RIF1, a versatile protein pivotal in driving antibody isotype diversification by safeguarding DNA ends during class switch recombination (CSR). This manuscript described a notable consequence of RIF1 deficiency: a marked increase in plasmablast (PB) formation under ex vivo conditions, coupled with an expedited transition to plasma cells (PCs) upon immunization. Intriguingly, these observations unfolded independently of RIF1's traditional roles in DNA repair and CSR mechanisms. Instead, this came from RIF1's capacity to modulate the transcriptional activity of specific BLIMP1 target genes. Hence, beyond its recognized function in augmenting antibody diversity, RIF1 potentially emerges as a critical regulator, fine-tuning the temporal dynamics of late B cell differentiation and adding yet another layer of intricacy to the orchestration of humoral immunity. However there are few major points that need to be addressed prior to further consideration.

1. Data are needed to prove that immature B cells from the Rif1F/FCd19Cre/+ mice were RIF1 deletion.
2. In line 151, "mice knock-out for AID exhibited the expected massive increase in GC B cells and a cellular distribution heavily skewed towards the LZ", but the DZ/LZ ratio was much less. Please check it.
3. In line 154, It was described that Rif1F/FCd19Cre/+ mice displayed near-physiological levels of IgA+ GC B cells while there is statistically significant difference in Figure 3b. Please check the statistics since the percentage of IgA+ GC B cells did not seem to be statistically different.
4. For Figure 4b, the mice numbers were different in the number of TACI+ CD138+ cells in BM.
5. Luciferase assays should be carried out to see if RIF1 directly influences BLIMP1 expression.
6. MAD2L2 expression also varied significantly in the different B cell subtypes. And this result should be discussed since 53BP1 and Shieldin 1 showed no difference.

Reviewer #4

(Remarks to the Author)

I co-reviewed this manuscript with one of the reviewers who provided the listed reports. This is part of the Nature

Communications initiative to facilitate training in peer review and to provide appropriate recognition for Early Career Researchers who co-review manuscripts.

Version 1:

Reviewer comments:

Reviewer #1

(Remarks to the Author)

The authors have addressed my queries and/or attempted to do so. I suggest publication of this study.

Reviewer #2

(Remarks to the Author)

The authors have carefully addressed my concerns. i have no further issues, and recommend publication.

Reviewer #3

(Remarks to the Author)

The authors have addressed my concerns.

Reviewer #4

(Remarks to the Author)

We thank the Editor and all Reviewers for the interest and overall positive response expressed in reference to our study, as well as for the constructive criticisms that have helped to further improve our manuscript. Please find below our detailed point-by-point response to their comments.

Reviewer #1 (Remarks to the Author):

RahjoueiDiVirgilio et al.

A series of programmed differentiation steps driven by various transcription factors (TFs) orchestrates the development of common lymphoid progenitor (CLPs) to antibody secreting B cells (plasmablasts (PB) or plasma cells (PC)). Various transcription factors and other transcriptional regulatory events drive each of the developmental steps, during early B cell development in the bone marrow, in the germinal center and finally during ASC differentiation in the periphery. RIF1 protein has been previously shown to be important for prevention of resection of DNA double strand breaks and promote NHEJ during class switch recombination, a DNA recombination step that defines the constant region of antibodies expressed by ASCs. In this manuscript, the authors aim to establish that in addition to DNA break end repair/resection suppression activity, RIF1 also functions directly as a transcription regulatory factor by binding to BLIMP1 sites, a TF that promotes APC differentiation, and thereby regulates the kinetics of APC development.

The authors have nicely demonstrated that B cells increase RIF1 expression during late B cell differentiation (Fig. 1). RIF1 depletion leads to increased kinetics of APC differentiation in vivo and ex vivo (Fig. 2) without affecting germinal center dynamics (Fig. 3). In vivo, immunization of mice deficient of RIF1 leads to increased level of plasma B cell formation (Fig. 4). Finally, a series of experiments evaluating genomic occupancy of RIF1 using ChIP experiments indicates that RIF1 overlaps BLIMP binding sites and therefore antagonizes BLIMP function to enhance ASC/PC differentiation (Fig. 6 and 7).

Overall, I found the data quite important, compelling and the manuscript very well written. Since it is assumed that RIF1 functions in regulating DNA resection, its functions as a transcription regulator is a surprise and can be considered as a novel discovery. Thus, I am supportive of publication of this study. Below I list a few questions that the authors could consider addressing during revisions.

We thank the Reviewer for finding our data “quite important, compelling” and for supporting the publication of our study. We greatly appreciate the constructive comments that she/he raised, and have included our response to each question below.

1. What is the effect of RIF1 over expression on in vitro cultured GC B cells in terms of APC differentiation. If the experiment in Fig. 2d and 2e is performed with RIF1 over expressor cells, will that attenuate CD138+ B cell levels.

We greatly appreciated this question and dedicated considerable experimental effort to developing and testing a suitable approach to address it. While primary B lymphocytes are resistant to traditional transfection methods, they can be efficiently transduced. However,

the retroviral packaging limit represents a constraint to the delivery of large constructs, which is unfortunately the case for RIF1 (2426 amino acids). To bypass this technical limitation, we implemented CRISPR activation (CRISPRa) in primary B cell cultures (new Fig. S6). To do so, we derived the *Rosa26^{dCas9-SunTag/+}* mouse model from *Gt(ROSA)26Sor^{tm1(CAG-cas9*,-BFP)Khk}*/Mmjax mice (RRID: MMRRC_043926-JAX, ¹), and optimized the transduction of B cells with a single-vector construct that expresses both scFv fused to VP64 and a promoter-targeting gRNA (new Fig. S6a). This approach enabled the successful overexpression of RIF1 in *ex vivo* activated B cells, achieving levels up to 10 times higher than physiological expression (new Fig. S6, b and e).

[panel redacted]

Figure for Reviewers R1. Assessment of the iGB system performance in combination with the *ex vivo* CRISPRa protocol. **a** Schematic representation of the integration of the CRISPRa protocol with the iGB system. d: day; α CD40: anti-CD40 antibody; CD40L: CD40 ligand; TSS: transcription start site. **b** Flow cytometry plots displaying the kinetics of transduced (GFP+) cell loss in one sample per condition. The plots are representative of six independently infected primary cultures per condition. **c** Graph summarizing the percentage of CD138⁺ cells in the indicated cell population at day 6 of the cell culture system shown in panel a. Data is representative of four (EV) and six (all other samples) repeats in three independent experiments. Uninf: uninfected culture; EV: empty vector. Significance in panel c was calculated with one-way Anova test (multiple comparisons) and error bars represent SD. ns: not significant; **** = $p \leq 0.0001$.

The efficiency of *ex vivo* plasmablast (PB) differentiation positively correlates with the duration of primary B cell culture, and is therefore adversely affected by the infection protocol (data not shown). To bypass this limitation, we combined the CRISPRa approach with the induced GC B (iGB) cell culture system ² (Fig. R1a), which considerably extends time and dynamic range for differentiation (Fig. 2). However, we observed that infected B cells were rapidly counter-selected in the iGB system, regardless of the transduced construct (Fig. R1b). As a result, the experimental window was shortened from four to just two days of IL-21-induced differentiation. In addition, the infected cultures exhibited a markedly reduced differentiation potential (Fig. R1c, compare uninfected samples, Uninf, with empty vector, EV, and *gRandom* controls). These findings were consistently observed across experimental replicates (n = 6 mice in three independent experiments), and only the overexpression of BLIMP1 to levels exceeding controls by more than an order of magnitude (new Fig. S6, c and f) rescued CD138⁺ cell formation (Fig. R1c).

Under the experimental conditions described above, RIF1 overexpression did not further reduce CD138⁺ cell formation below the levels observed in empty-vector control (Fig. R1c). However, the combined effects of counter-selection of infected cells and the reduced differentiation potential prevent us from considering this result as a conclusive answer to the reviewer question, and are therefore provided as a Figure for Reviewers. Nonetheless, the CRISPRa protocol that we established overcomes a significant barrier to studying large proteins in primary B lymphocytes, making it an invaluable tool for mechanistic investigations in this model system. Specifically, in our study, it allowed us to demonstrate that RIF1 overexpression does not affect *Prdm1* expression (new Fig. S6d). This finding offers further support to our proposed mechanistic model (as detailed in our main response to Reviewer #2) and has been included in the revised manuscript as a Supplementary Figure (new Fig. S6).

2. Will RIF1F/F B cells accumulate genetic alterations or gene expression changes seen enriched in myeloma B cells.

Multiple myeloma (MM) is a slow-progressing disease usually seen in the elderly, with primary and secondary genetic “hits” accumulating over time. In addition, it is extremely heterogeneous in nature, and several MM subtypes have been identified in patients that differ not only in terms of tumor biology but also gene expression profile^{3,4}. This feature combination has considerably complicated the generation of preclinical MM models that faithfully recapitulate the genetic and transcriptional makeup of MM. In mice, the disease manifestation usually appears only in aged cohorts^{5–8}, and preferentially with the enforced expression of strong oncogenes or chromosomal translocations specifically observed in MM patients^{7,8}. Even under these conditions, the mouse tumors do not fully mimic their human counterparts. Therefore, despite the accelerated PC phenotype that we reported in *Rif1^{F/F}Cd19^{Cre/+}* mice, we expect the penetrance of MM genetic and/or gene expression alterations in the plasma cell compartment to be negligible, if present at all, would the mice be allowed to age considerably.

3. Is there a difference in kinetics of PC differentiation of switched and unswitched cells obtained from B cells that are depleted of RIF1 (RIF1F/F. CD19Cre).

To answer the Reviewer’s question, we took advantage of the iGB cell culture system, and monitored the IgM⁺ versus IgG1⁺ distribution of CD138⁺ populations over time during IL-21-driven differentiation (new panel g in Fig. 2). At the first assessment point (two days of IL-21 incubation and day 6 of the iGB culture), the isotype distribution of CD138⁺ cells closely mirrored the genotype-dependent CSR efficiencies observed in the IL-4-cultured conditions (Fig. 2, compare panels e and new panel g). During the final two days in culture, the proportion of IgG1⁺ cells increased in both groups, with IgM⁺ cells eventually disappearing from the CD138⁺ pools (new Fig. 2g). This finding is particularly striking in the case of *Rif1^{F/F}Cd19^{Cre/+}* cultures, which started with considerably higher levels of IgM⁺ cells than controls because of their CSR defect.

We conclude that, although unswitched and switched cells display similar differentiation potential, IgM⁺ PC-like cells are outcompeted by the switched counterparts in the iGB system. Importantly, the observation that RIF1-deficient B cells still show enhanced

differentiation supports our conclusion that this phenotype is independent from their CSR defect.

4. In Fig. 7e, does the occupancy of BLIMP1 increase at target genes following RIF1 depletion. How are DNA breaks induced RIF1 rpeaks separated from those peaks recruited for APC differentiation (BLIMP1 antagonistic peaks). Is that possible to do.

We thank the Reviewer for giving us the opportunity to discuss the DNA damage-dependent *versus* independent recruitment of RIF1 to the activated B cell genome.

To address whether BLIMP1 occupancy at its target genes increases following RIF1 depletion, we would need to perform BLIMP1 ChIP-Seq/-qPCR in B cells from *Rif1^{F/F}Cd19^{Cre/+}* mice. This approach is presently unfeasible because functional ChIP-grade antibodies specific to mouse BLIMP1 (whether commercial or in-house) are currently unavailable. Meinrad Busslinger (IMP, Vienna) and Stephen Nutt (WEHI, Victoria, Australia), whose laboratories have characterized BLIMP1 genome binding profile in ASC differentiation^{9,10}, have both confirmed this information (personal communication). Indeed, to identify BLIMP1 target genes, the Busslinger group intentionally generated a mouse model, *Prdm1^{Bio/Bio}Rosa26^{BirA/+}*, which enables the streptavidin-mediated precipitation of BLIMP1 fused to a biotin-acceptor motif (BLIMP1-Bio-ChIP-seq)¹⁰. To repeat this assay in the absence of RIF1, we would need to import *Prdm1^{Bio/Bio}Rosa26^{BirA/+}* mice (currently only available as cryo-preserved material as per recent communication with M. Busslinger) and breed them with our *Rif1^{F/F}Cd19^{Cre/+}* mice. Such a combination of (four) alleles would require a very long, costly, and labor-intensive breeding plan, which is unfortunately also incompatible with the revision timeline.

Nonetheless, we can confidently state that the recruitment of RIF1 to BLIMP1 target genes is independent from DNA damage.

In activated B cells, DSBs arise stochastically as byproducts of cellular metabolism (including unperturbed DNA replication), and at defined genomic locations in response to AID activity (S regions at *Igh* and off-targets) and replication stress (common fragile sites – CFS – and early replicating fragile sites – ERFs). The stringent p-value threshold of 10^{-6} that we applied for RIF1 peak selection ensures that the distribution of reads in specific regions is highly unlikely to be random, with a probability of 1 in 1,000,000. Accordingly, all RIF1 peaks selected for validation were reproducibly confirmed by the ChIP-qPCR experiments performed for this revision (new Figures 6d and 8d). The stringent peak calling criterion implies a high confidence that the identified peaks do not represent stochastic DSB events, since genomic breaks occurring randomly in individual cells would not lead to a consistent enrichment across repeated observations in bulk data. Even reproducible DSBs at specific genomic sites would generate RIF1 peaks only if occurring synchronously in a sizable portion of the culture. In agreement with this point, we found that only a negligible fraction of regions containing both RIF1 and BLIMP1 peaks overlapped with AID off-target sites (new Fig. 7c). Furthermore, ERFs accounted for just 10% of these regions (new Fig. 7d), despite ERFs being very large and covering sizable portions of the genome¹¹. More importantly, none of the co-regulated repressed targets localized close to ERFs and only one was in proximity of an AID off-target hotspot. CFSs were not included in our analysis since only eight such regions have been identified in mice¹².

Altogether, these analyses indicate that RIF1 association with BLIMP1 target genes is not a result of DNA damage-dependent recruitment. We have now incorporated these findings into the revised manuscript.

5. Overall, the manuscript is very nicely written. The discussion could cover a few more points.

- (1) At a molecular and biophysical level how are BLIMP1 and RIF1 competing. The peaks shown in Fig 7 are overlapping partially but not completely. Is there a specific RIF1 binding motif that overlaps with BLIMP1 binding motif identified from the RIF1/BLIMP1 peak calling? Is there another explanation.

As the Reviewer correctly pointed out, RIF1 exhibits multiple modalities of interaction with BLIMP1 target genes (current Fig. 8c, previously Fig. 7e). Motif enrichment analysis of co-occupied regions identified BLIMP1 binding site as the top-ranked motif (new Table S6). However, RIF1 peaks are quite broad (new Fig. 7a) and contain consensus binding motifs for a high number of transcription factors (new Fig. 6e and new Table S5). This observation indicates that RIF1 and BLIMP1 might exert their gene co-regulatory function not only by competing for binding to the same DNA regions but also by interacting with distinct cis-regulatory elements. Our proposed mechanism is that RIF1 gets recruited to the chromatin, and in turn modulates gene expression, in association with different transcriptional regulators. In support of this idea, characterization of RIF1 interactome in activated B cells identified several transcription factors as potential RIF1 interactors¹³.

We have now included this discussion point, the *de novo* motif discovery analysis (MEME-ChIP) of RIF1 peaks, and the enrichment motif analysis (SEA) of RIF1 and BLIMP1 co-occupied regions in the revised manuscript (Results and Discussion sections).

- (2) The aspects of AID mutations on single strand DNA to generate DNA DSBs, how that occurs and how RIF1 controls DNA repair at AID induce DSBs could be covered better. The mechanism of AID, RIF1 and CSR is not discussed properly for a complete understanding of RIF1 function in CSR versus APC differentiation.

To aid in the understanding of the DNA end processing vs APC differentiation regulatory functions of RIF1, we have now included extended paragraphs in the revised manuscript on the mechanisms of AID-dependent formation of DNA breaks and RIF1's involvement in their repair. Since both mechanisms are well established, we have opted to include this information in the Introduction rather than in the Discussion.

- (3) Finally, the authors mention that RIF1 can alter chromatin accessibility and target gene expression via its interaction with Polycomb group proteins. Chromatin accessibility is dependent upon basal noncoding RNA expression and on various epigenetic marks, among other factors. Outlining RIF1 collaboration with ncRNA biology and/or epigenetic marks, if possible, will make the discussion interesting (and substantially different than what had already been stated in the introduction

and results sections).

We thank the Reviewer for suggesting this interesting discussion point. We have now more explicitly highlighted how the interplay between RIF1, KAP1, and the *in cis*-acting lncRNAs *Xist* and *Tsix* exemplifies: 1) RIF1's role as a direct and positive regulator of transcription; and 2) its critical contribution to chromatin state modulation during a key stage of embryonic stem cell differentiation. Hence, our findings and proposed model in activated B cells support the existence of widespread mechanisms for RIF1-dependent transcriptional modulation across cell types and differentiation states.

Reviewer #2 (Remarks to the Author):

The manuscript by Di Virgilio examines the role of RIF1 in B cell biology. This factor is known to be involved in DNA end protection to facilitate NHEJ. But other literature suggests that it has other functions, including activities in modulating transcriptional networks. In this report, the authors show that RIF1's expression is increased in stimulated B cells. They further show that plasma cells (CD138+) are increased by ~2-fold in RIF KO cells without an effect on GC dynamics, with the exception of showing the expected effect on isotype switching. The authors further show that RIF KO B cells have about a 2-fold increase in Blimp1 (*Prdm1*) expression, and that RIF binds to genes involved in the adaptive immune response, and overlaps at many loci also bound by BLIMP1. The authors suggest that RIF1 binds to targets and inhibits BLIMP1 action there, although this last section was not written clearly, so it is hard to interpret the data.

Overall, this is an interesting study, but the overall effects observed in the RIF1 KO were not striking. The manuscript would benefit from a more detailed analysis of RIF1 binding to specific genetic regions (and to the *Prdm1* gene, itself) and how this binding modulates BLIMP1 function, as well as BLIMP1 expression since Blimp1 expression is increased in the RIF KO cells.

We thank the Reviewer for highlighting that "this is an interesting study" and for providing excellent suggestions on how to make our work more compelling.

In response to the reviewer's request, we further investigated the mechanisms through which RIF1 modulates BLIMP1-dependent ASC differentiation by first examining RIF1 binding to *Prdm1*. Our RIF1 ChIP-Seq data identified two peaks located tens of kilobases upstream of the canonical transcriptional start site (Fig. R2), and we confirmed RIF1 binding to these regions *via* ChIP-qPCR (Fig. R2). However, we found no evidence of functional impact. As part of this revision, we successfully implemented CRISPRa in primary B cell cultures and demonstrated that overexpression of RIF1 in *ex vivo* activated B cells did not alter BLIMP1 transcript levels (new Fig. S6). This finding complements a previous study, which showed that shRNA-mediated downregulation of RIF1 in the mouse B cell lymphoma line BAL17 did not affect BLIMP1 transcript levels¹⁴. While we acknowledge the negative nature of these results, the lack of functional evidence across approaches strongly suggests that RIF1 does not directly regulate BLIMP1 expression.

More importantly, the substantial overlap of RIF1 occupancy at BLIMP1 targets that are also differentially regulated in the absence of RIF1 is predominantly observed at a subset of BLIMP1 repressed genes (current Fig. 7, a and b). In contrast, the overlap at upregulated BLIMP1 target genes is much more limited and only becomes apparent at late time points (current Fig. 8b and Table S7). If RIF1's ability to counteract BLIMP1-mediated ASC differentiation were indirectly due to regulation of BLIMP1 expression, we would expect RIF1 depletion to equally affect both activated and repressed BLIMP1 targets, and with the same kinetics. However, this is not the case.

Finally, we have validated the results of RIF1 ChIP-Seq at representative repressed BLIMP1 targets that are regulated by RIF1, and confirmed the binding of RIF1 to all selected loci (new Fig. 8d).

Taken all together, our data supports a model in which RIF1 modulates PC differentiation by directly regulating the transcriptional status of a subset of BLIMP1-repressed targets, rather than by altering BLIMP1 expression itself.

Please find our responses to the additional comments below, including those addressing the strength of the PC phenotype and more detailed information on the analysis of RIF1 binding to specific genomic regions. In addition, we have reworded the last part of the Results section to facilitate the interpretation of the corresponding findings and proposed mechanism.

Additional comments

1. The flow cytometry analysis in Fig 4b does not show much of an increase in PCs as a result of RIF1-deletion, as suggested by the authors. In addition, the absolute numbers of cells needs to be reported here and in previous figures (including supplemental figures) as the total cellularity may have been affected in the RIF1 KO mice that may give the impression that PC formation is increased in the absence of RIF1.

We agree with the Reviewer that the increase in PC levels in the spleen and bone marrow of *Rif1^{F/F}Cd19^{Cre/+}* mice cannot be described in term of "fold-change". Nonetheless, the median increase over control (*Cd19^{Cre/+}*) totals up to 39-50% more PCs in the spleen of *Rif1^{F/F}Cd19^{Cre/+}* mice after immunization (Fig. 4b). These values are actually quite substantial, especially if we take into account that the phenotypic analysis assesses the total PC pool rather than zeroing in on the antigen-specific fraction. Even when considering the more limited increase in the bone marrow (median PC increase of 17%, Fig. 4b), the phenotype is highly reproducible (described as "consistent increase" in the text) and it is also corroborated by the ELISpot analysis, which provides information on both number and function of NP-specific PCs (Fig. 4c). Given the immune system's resilience, the fact that we were able to recapitulate *in vivo* the observations from our *ex vivo* model systems and the phenotype robustness were both unexpected and positively surprising findings.

We have now also provided the total spleen and bone marrow cellularity for all mice employed in the phenotypic characterization of the PC compartments (new graphs in Fig. S3 and S4). The newly-incorporated data shows that the number of cells retrieved from the

spleens and bone marrows of *Rif1^{F/F}Cd19^{Cre/+}* mice is comparable to controls, thus indicating that the increased percentage of PCs observed in immunized *Rif1^{F/F}Cd19^{Cre/+}* mice cannot be explained by differences in total cellularity.

Indeed, we believe that the combination of multiple *ex vivo* assays and the thorough *in vivo* analysis represents one of the main strengths of our study, as it unambiguously supports our conclusion of a novel cell-intrinsic RIF1 ability to modulate terminal B cell differentiation.

For improved clarity, we have now included the median PC increase values from the phenotypic analysis and the reference to the total cellularity data in the corresponding paragraphs of the Result section.

2. Line 119: Not clear why the authors stated this. To show this effect on increased PB and how it is not related to end protection, the authors need to do double KOs with RIF1 and AID, for example.

Work performed in the last decade in the Durocher, Boulton, de Lange, Jackson, Chapman, Nussenzweig, Deriano, our and many other DNA Repair labs has collectively and carefully dissected the DSB end protection machinery. RIF1 is the first factor in the 53BP1-initiated cascade, and it protects DSB ends against nucleolytic resection *via* its ability to recruit the key downstream effector Shieldin, which is a protein complex comprising MAD2L2, SHLD1, SHLD2, and SHLD3 subunits^{13,15–28}. Deficiency in any of these subunits abrogates RIF1-dependent DSB end protection, thus leading to a severe CSR defect^{19–25,29,30}.

To unambiguously assess whether the phenotype we observed in *Rif1^{F/F}Cd19^{Cre/+}* mice is caused by their defective DSB end protection capability, we had purposely imported *Shld1^{-/-}* mice from the Deriano lab (Institut Pasteur, Paris) to be analysed in parallel. This genotype provides the ideal separation-of-function setup to specifically abrogate RIF1-dependent DSB end protection without affecting other known or yet-to-be characterized RIF1 activities. In contrast, deletion of AID in RIF1-deficient B cells would only prevent the formation of DSBs (predominantly) at the *Igh* locus, but would not address the DNA end protection aspect. Therefore, while we highly value the Reviewer's inquiry for clarification, we still believe that the fact that the increased PB formation is observed in RIF1-, but not Shieldin-deficient B cells, conclusively proves that the newly-reported phenotype is independent from DSB end protection (Fig. 2c). As an additional note, the single-AID-knock-out condition (*Aicda^{-/-}* mice) was included in the analysis as a secondary control to show that increased PB formation is not merely a consequence of defective CSR *per se*.

For improved clarity, we have now included the above-stated explanation in the corresponding paragraph of the Result section, and the graph with the CSR efficiencies of the same cultures used to assess PB formation in the figure (modified Fig. 2c). In addition, we have also better elaborated on RIF1-mediated DSB end protection cascade in the revised manuscript.

3. The RNAseq experiments shown in Fig 5 suggest that many genes are differentially regulated in the KO B cells, but in the text they suggest only a few genes (line 205). What is the discrepancy here? Also, the authors checked for expression of B cell identity genes (Fig

5c): were these selected based on the RNAseq? There was not difference in expression in most genes examined which may or may not disagree with the RNAseq data.

We agree with the Reviewer that the wording we used to describe the results of the RNA-Seq analysis (Fig. 5) may give the impression of a discrepancy in our data interpretation. When we wrote that "Comparative assessment of the transcriptional profiles of *Cd19^{Cre/+}* and *Rif1^{F/F}Cd19^{Cre/+}* B cells identified only a limited number of considerably deregulated genes in the absence of RIF1 (N° genes with log₂ FC < -1 and > 1 = 0, 105, and 47 at 48, 72, and 96 h, respectively)", we meant to highlight that, even if many genes are differentially regulated in *Rif1^{F/F}Cd19^{Cre/+}* B cells from a statistical perspective (149, 732, 672 up- and 230, 662, 981 down-regulated as per adjusted p-value ≤ 0.05 at 48, 72, and 96 h post-activation, respectively; current Fig. S5b and Table S1), only a limited number of them exhibits a considerable change in expression levels in the absence of RIF1 (log₂ FC < -1 and > 1; Table S1). We have now better explained this point in the revised manuscript text and in Figure 5 (modified panel b).

Given our aim to explain the enhanced plasma cell phenotype observed in RIF1-deficient B cells, one plausible hypothesis was that key transcriptional regulators of mature B cell identity (*Pax5*, *Ebf1*, *Foxo1*, and *Bach2*) or key factors driving the antibody secreting cell (ASC) program (*Prdm1*, *Irf4*, and *Xbp1*) may be differentially regulated in the absence of RIF1. For this reason, and as indicated in the legend to Fig. 5, in panel c we zoomed in on the RNA-Seq analysis from panels a and b to derive the relative expression levels of this set of genes. As the Reviewer correctly pointed out, none of these genes was differentially regulated in the absence of RIF1, with the exception of *Prdm1*, which exhibited significantly higher levels of expression in *Rif1^{F/F}Cd19^{Cre/+}* B cells at 96 h post-activation, albeit with a log₂ FC < 1 (also shown in panel 5b and Table S1).

4. The CHIPseq data needs validation, at a minimum at a few loci.

As requested by the reviewer, we conducted validation by ChIP-qPCR for multiple genomic regions containing RIF1 peaks identified in our RIF1 ChIP-Seq analysis. The selected regions comprised a combination of RIF1 peaks in genes categorized under "Regulation of Adaptive Immune Responses" in the gene ontology enrichment analysis (Fig. 6c) as well as representative repressed BLIMP1 targets from the subset that we showed to be regulated by RIF1. To enhance the rigor of our validation: 1) we performed three independent RIF1 ChIP-qPCR experiments, using two mice per genotype (*WT* for mock ChIP and *Rif1^{FH/FH}* for RIF1 ChIP) in each experiment, for a total of six mice per group; 2) two different operators carried out these experiments; and 3) we included regions lacking RIF1 peaks as additional negative controls in parallel to mock ChIP samples from *WT* mice. The results confirmed the presence of RIF1 at all selected regions, and the data has been included in the revised manuscript (new Figures 6d and 8d).

Reviewer #3 (Remarks to the Author):

The establishment of protective immune responses hinges upon the robust capability of mature B cells to secrete a diverse array of antigen-specific antibodies, each endowed with distinct effector functions. At the heart of this process lies RIF1, a versatile protein pivotal in

driving antibody isotype diversification by safeguarding DNA ends during class switch recombination (CSR). This manuscript described a notable consequence of RIF1 deficiency: a marked increase in plasmablast (PB) formation under *ex vivo* conditions, coupled with an expedited transition to plasma cells (PCs) upon immunization. Intriguingly, these observations unfolded independently of RIF1's traditional roles in DNA repair and CSR mechanisms. Instead, this came from RIF1's capacity to modulate the transcriptional activity of specific BLIMP1 target genes. Hence, beyond its recognized function in augmenting antibody diversity, RIF1 potentially emerges as a critical regulator, fine-tuning the temporal dynamics of late B cell differentiation and adding yet another layer of intricacy to the orchestration of humoral immunity. However there are few major points that need to be addressed prior to further consideration.

We are thankful to the Reviewer for stating that “This manuscript described a notable consequence of RIF1 deficiency” and that “beyond its recognized function in augmenting antibody diversity, RIF1 potentially emerges as a critical regulator, fine-tuning the temporal dynamics of late B cell differentiation and adding yet another layer of intricacy to the orchestration of humoral immunity”. Please find below our responses and the explanations to clarify all points that the Reviewer brought to our attention.

1. Data are needed to prove that immature B cells from the *Rif1^{F/F}Cd19^{Cre/+}* mice were RIF1 deletion.

To address the Reviewer's point, we have assessed the status of the *Rif1* conditional allele in *Rif1^{F/F}Cd19^{Cre/+}* mice across key B cell developmental stages. Based on the transcriptional control of the B lineage-restricted *Cd19* gene, Cre expression in *Cd19^{Cre/+}* mice is expected to start at the pre-B cell stage and to increase as cells mature³¹. Accordingly, we observed that Cre-mediated deletion of the *Rif1* floxed allele is evident only in a fraction of the Pre-Pro B cell population (bone marrow-derived B220⁺IgM⁻); however, it occurs in the vast majority of immature B cells (BM B220⁺IgM⁺). The deletion is near-complete in the periphery (splenic B220⁺IgM⁺), which is in agreement with our previous report showing that *Rif1* floxed allele is undetectable in *ex vivo* stimulated splenic B cells from *Rif1^{F/F}Cd19^{Cre/+}* mice¹³. The minor fraction of immature B cells bearing an intact *Rif1^F* allele would lead, if persisting *in vivo*, to an underestimation of the increased plasma cells phenotype that we observed in these mice, and would therefore only strengthen the conclusions of our study. The data is now summarized in a new supplementary figure in the revised manuscript (new Fig. S2).

2. In line 151, “mice knock-out for AID exhibited the expected massive increase in GC B cells and a cellular distribution heavily skewed towards the LZ”, but the DZ/LZ ratio was much less. Please check it.

Previous work from the Martin lab showed that *Aicda^{-/-}* mice display enlarged GCs as result of the combined effect of reduced susceptibility to apoptosis and inefficient differentiation to plasma cells^{32,33}. Micro-anatomically, AID-deficient GC B cells are present in physiological numbers as centroblasts in the dark zone (DZ), but accumulate as centrocytes in the light zone (LZ)^{32,34}. As a consequence, the relative distribution of GC B cells is heavily skewed towards the LZ in *Aicda^{-/-}* mice, which indeed leads to a ratio between DZ and LZ B cells smaller than in wild-type mice. Given these well-characterized phenotypes, we employed

Aicda^{-/-} mice in the GC dynamics assessment as the positive control for increased number of GC B cells and altered DZ-to-LZ distribution, which we both faithfully recapitulated.

3. In line 154, It was described that *Rif1*^{F/F}*Cd19*^{Cre/+} mice displayed near-physiological levels of IgA⁺ GC B cells while there is statistically significant difference in Figure 3b. Please check the statistics since the percentage of IgA⁺ GC B cells did not seem to be statistically different.

The significance between the two groups was calculated using the Mann–Whitney U test and indicates an actual difference between median values of -5.8 (*Cd19*^{Cre/+} median = 52.3 for n = 7 values, and *Rif1*^{F/F}*Cd19*^{Cre/+} median = 46.5 for n=7 values) with an exact p value of 0.0379, represented with one star in the graph (* = p ≤ 0.05). We can therefore confirm that the levels of IgA⁺ GC B cells in controls and *Rif1*^{F/F}*Cd19*^{Cre/+} mice are statistically different. However, the difference is modest, which is particularly striking considering the intrinsic CSR defect of RIF1-deficient B cells^{13,16,17}. This is the reason why we described the levels of CSR in *Rif1*^{F/F}*Cd19*^{Cre/+} Payer's Patches as near-physiological and drew attention to this result in the manuscript text.

4. For Figure 4b, the mice numbers were different in the number of TACI⁺ CD138⁺ cells in BM.

This was a mistake on our side, caused by the erroneous generation of this graph from a previous version of the experiments' summary file that did not include all of the experimental repeats. As correctly indicated in the figure legend, the graphs summarize the results of at least six mice per genotype and time point in at least 3 independent experiments, with the following specifications:

Day 7 - n = six *Cd19*^{Cre/+} and n= nine *Rif1*^{F/F}*Cd19*^{Cre/+} mice

Day 14 - n = six *Cd19*^{Cre/+} and n = seven *Rif1*^{F/F}*Cd19*^{Cre/+} mice

Day 28 - n = eight *Cd19*^{Cre/+} and n = seven *Rif1*^{F/F}*Cd19*^{Cre/+} mice

We apologize and thank the Reviewer for catching the mistake. We have now re-verified all of the graphs in the panel and corrected the figure accordingly (modified Fig. 4b).

5. Luciferase assays should be carried out to see if RIF1 directly influences BLIMP1 expression.

Luciferase assays are a well-established method in transient transfection experiments. However, this approach is not suitable for primary B cell cultures, which heavily rely on retroviral transduction. To our knowledge, there are no commercially available viral vectors designed for dual-luciferase assays, which are essential for normalizing signal measurements across samples with varying infection efficiency. To address this issue, we initiated the development of Firefly-Renilla luciferase retroviral vectors from scratch. During the revision period, we generated two iterations of dual-luciferase constructs and tested four different promoters. Despite these efforts, to date, we have been unable to obtain optimal luciferase expression in B cells to enable the assay.

We would however like to point out that, even if we were able to overcome these technical limitations, we would still expect a negative result. Previous work from Morgan *et al.*³⁵ showed that *Prdm1* exhibits multiple promoter usage depending on the cell type. Our Chip-Seq analysis in activated B cells identified two distinct genomic regions particularly enriched with RIF1 at the *Prdm1* locus, which we have now validated by CHIP-qPCR (Fig. R2).

However, neither of these regions corresponds to the promoter driving *Prdm1* expression from the canonical transcriptional start site (TSS) (Fig. R2). Of note, region 2 is located 70 Kb upstream the TSS, near or overlapping with an alternative distal promoter identified by Morgan *et al.* in the mouse yolk sac³⁵. This distal element is, however, dispensable for *Prdm1* expression since, in contrast to the canonical promoter, its deletion does not affect BLIMP1 levels and antibody secretion³⁵.

Figure for Reviewer R2. Validation of *Rif1* occupancy at the *Prdm1* locus in activated B cells. Graph depicting RIF1 occupancy as determined by ChIP-qPCR at two regions (reg) upstream *Prdm1* canonical transcriptional start site (TSS). The graph summarizes data from four mice per genotype in two independent experiments. The inset over the graph shows the RIF1 ChIP-Seq tracks from *WT* and *Rif1^{FH/FH}* B cells at the locus. The orange boxes delineate RIF1 peaks as determined by RIF1 ChIP-seq. Significance was calculated with the Mann–Whitney U test and error bars represent SD. * = $p \leq 0.05$.

6. MAD2L2 expression also varied significantly in the different B cell subtypes. And this result should be discussed since 53BP1 and Shieldin 1 showed no difference.

We agree with the Reviewer that *Mad2l2* is the only other DNA end protection factor significantly upregulated in *ex vivo* activated B cells (our RNA-Seq, Fig. S1c). One possible explanation is that B cells upregulate *Mad2l2* after activation to support the increased demand for translesion DNA synthesis during proliferation and somatic hypermutation^{36,37}. However, given the minimal changes in *Mad2l2* expression across B cell lineage developmental stages (ImmGen Skyline analysis, Fig. S1a) and the absence of further experimental follow-up, we believe this hypothesis to remain highly speculative. Therefore, while we appreciate the Reviewer's suggestion, we have respectfully chosen not to include this discussion point in the revised manuscript.

Reviewer #4 (Remarks to the Author):

We recognize that peer-reviewing is a demanding and time-consuming task, yet it is crucial to guarantee the quality of scientific publishing. We are very thankful to this (and the other)

Reviewer(s) for the time and efforts invested into assessing/evaluating our manuscript and providing constructive feedback.

References

1. Wangensteen, K. J. *et al.* Combinatorial genetics in liver repopulation and carcinogenesis with a novel in vivo CRISPR activation platform. *Hepatology* **68**, 663–676 (2018).
2. Nojima, T. *et al.* In-vitro derived germinal centre B cells differentially generate memory B or plasma cells in vivo. *Nature Communications* **2**, 1–11 (2011).
3. Zhan, F. *et al.* The molecular classification of multiple myeloma. *Blood* **108**, 2020–2028 (2006).
4. Broyl, A. *et al.* Gene expression profiling for molecular classification of multiple myeloma in newly diagnosed patients. *Blood* **116**, 2543–2553 (2010).
5. Suematsu, S. *et al.* Generation of plasmacytomas with the chromosomal translocation t(12;15) in interleukin 6 transgenic mice. *Proceedings of the National Academy of Sciences* **89**, 232–235 (1992).
6. Scherger, A. K. *et al.* Activated gp130 signaling selectively targets B cell differentiation to induce mature lymphoma and plasmacytoma. *JCI Insight* **4**, e128435, 128435 (2019).
7. Wen, Z. *et al.* Expression of NrasQ61R and MYC transgene in germinal center B cells induces a highly malignant multiple myeloma in mice. *Blood* **137**, 61–74 (2021).
8. Winkler, W. *et al.* Mouse models of human multiple myeloma subgroups. *Proceedings of the National Academy of Sciences* **120**, e2219439120 (2023).
9. Tellier, J. *et al.* Blimp-1 controls plasma cell function through the regulation of immunoglobulin secretion and the unfolded protein response. *Nat Immunol* **17**, 323–330 (2016).
10. Minnich, M. *et al.* Multifunctional role of the transcription factor Blimp-1 in coordinating plasma cell differentiation. *Nat Immunol* **17**, 331–343 (2016).
11. Barlow, J. H. *et al.* Identification of early replicating fragile sites that contribute to genome instability. *Cell* **152**, 620–632 (2013).
12. Helmrich, A., Stout-Weider, K., Hermann, K., Schrock, E. & Heiden, T. Common fragile sites are conserved features of human and mouse chromosomes and relate to large active genes. *Genome Res* **16**, 1222–1230 (2006).
13. Di Virgilio, M. *et al.* Rif1 Prevents Resection of DNA Breaks and Promotes Immunoglobulin Class Switching. *Science* **339**, 10.1126/science.1230624 (2013).
14. Tanaka, H. *et al.* Epigenetic Regulation of the Blimp-1 Gene (Prdm1) in B Cells Involves Bach2 and Histone Deacetylase 3. *The Journal of biological chemistry* **291**, 6316–6330 (2016).
15. Zimmermann, M., Lottersberger, F., Buonomo, S. B., Sfeir, A. & De Lange, T. 53BP1 regulates DSB repair using Rif1 to control 5' end resection. *Science* **339**, 700–704 (2013).
16. Chapman, J. R. *et al.* RIF1 Is Essential for 53BP1-Dependent Nonhomologous End Joining and Suppression of DNA Double-Strand Break Resection. *Molecular Cell* **49**, 858 (2013).
17. Escribano-Díaz, C. *et al.* A Cell Cycle-Dependent Regulatory Circuit Composed of 53BP1-RIF1 and BRCA1-CtIP Controls DNA Repair Pathway Choice. *Molecular Cell* **49**, 872–883 (2013).

18. Feng, L., Fong, K.-W., Wang, J., Wang, W. & Chen, J. RIF1 counteracts BRCA1-mediated end resection during DNA repair. *J Biol Chem* **288**, 11135–11143 (2013).
19. Xu, G. *et al.* REV7 counteracts DNA double-strand break resection and affects PARP inhibition. *Nature* **521**, 541–544 (2015).
20. Boersma, V. *et al.* MAD2L2 controls DNA repair at telomeres and DNA breaks by inhibiting 5' end resection. *Nature* **521**, 537–540 (2015).
21. Noordermeer, S. M. *et al.* The shieldin complex mediates 53BP1-dependent DNA repair. *Nature* **560**, 117–121 (2018).
22. Gupta, R. *et al.* DNA Repair Network Analysis Reveals Shieldin as a Key Regulator of NHEJ and PARP Inhibitor Sensitivity. *Cell* **173**, 972–988.e23 (2018).
23. Ghezraoui, H. *et al.* 53BP1 cooperation with the REV7-shieldin complex underpins DNA structure-specific NHEJ. *Nature* **560**, 122–127 (2018).
24. Dev, H. *et al.* Shieldin complex promotes DNA end-joining and counters homologous recombination in BRCA1-null cells. *Nat Cell Biol* **20**, 954–965 (2018).
25. Findlay, S. *et al.* SHLD2/FAM35A co-operates with REV7 to coordinate DNA double-strand break repair pathway choice. *EMBO J* **37**, e100158 (2018).
26. Gao, S. *et al.* An OB-fold complex controls the repair pathways for DNA double-strand breaks. *Nat Commun* **9**, 3925 (2018).
27. Tomida, J. *et al.* FAM35A associates with REV7 and modulates DNA damage responses of normal and BRCA1-defective cells. *EMBO J* **37**, e99543 (2018).
28. Mirman, Z. *et al.* 53BP1-RIF1-shieldin counteracts DSB resection through CST- and Pol α -dependent fill-in. *Nature* **560**, 112–116 (2018).
29. Ling, A. K. *et al.* SHLD2 promotes class switch recombination by preventing inactivating deletions within the Igh locus. *EMBO Rep* **21**, e49823 (2020).
30. Vincendeau, E. *et al.* SHLD1 is dispensable for 53BP1-dependent V(D)J recombination but critical for productive class switch recombination. *Nat Commun* **13**, 3707 (2022).
31. Rickert, R., Roes, J. & Rajewsky, K. B lymphocyte-specific, Cre-mediated mutagenesis in mice. *Nucleic acids research* **25**, 1317–1318 (1997).
32. Zaheen, A. *et al.* AID constrains germinal center size by rendering B cells susceptible to apoptosis. *Blood* **114**, 547–554 (2009).
33. Boulianne, B. *et al.* AID and Caspase 8 Shape the Germinal Center Response through Apoptosis. *The Journal of Immunology* **191**, 5840–5847 (2013).
34. Hogenbirk, M. A. *et al.* Differential Programming of B Cells in AID Deficient Mice. *PLOS ONE* **8**, e69815 (2013).
35. Morgan, M. A. J. *et al.* Blimp-1/Prdm1 Alternative Promoter Usage during Mouse Development and Plasma Cell Differentiation. *Molecular and Cellular Biology* **29**, 5813–5827 (2009).
36. Murakumo, Y. *et al.* A human REV7 homolog that interacts with the polymerase zeta catalytic subunit hREV3 and the spindle assembly checkpoint protein hMAD2. *J Biol Chem* **275**, 4391–4397 (2000).
37. Yang, D. *et al.* REV7 is required for processing AID initiated DNA lesions in activated B cells. *Nat Commun* **11**, 2812 (2020).